# Preserving ester-linked modifications reveals glutamate and aspartate mono-ADP-ribosylation by PARP1 and its reversal by PARG

Edoardo José Longarini [1,3] ✉ & Ivan Matić [1,2] ✉

Ester-linked post-translational modifications, including serine and threonine ubiquitination, have gained recognition as important cellular signals. However, their detection remains a significant challenge due to the chemical lability of the ester bond. This is the case even for long-known modifications, such as ADP-ribosylation on aspartate and glutamate, whose role in PARP1 signaling has recently been questioned. Here, we present easily implementable methods for preserving ester-linked modifications. When combined with a specific and sensitive modular antibody and mass spectrometry, these approaches reveal DNA damage-induced aspartate/glutamate mono-ADP-ribosylation. This previously elusive signal represents an initial wave of PARP1 signaling, contrasting with the more enduring nature of serine mono-ADP-ribosylation. Unexpectedly, we show that the poly-ADP-ribose hydrolase PARG is capable of reversing ester-linked mono-ADP-ribosylation in cells. Our methodology enables broad investigations of various ADP-ribosylation writers and, as illustrated here for noncanonical ubiquitination, it paves the way for exploring other emerging ester-linked modifications.

Post-translational modifications (PTMs) are key regulators of protein function and play important roles in virtually all cellular processes[1]. Our understanding of signaling pathways inevitably relies on the ability to identify and reliably detect such modifications. Routine sample preparation and analysis workflows often involve harsh physico-chemical conditions, such as high temperatures and extreme pH levels. These conditions can result in the artefactual loss of PTMs, thereby introducing systematic blind spots. Not surprisingly, most studies have historically focused on chemically stable PTMs, such as serine/threonine/tyrosine phosphorylation, lysine ubiquitination and lysine acetylation. In contrast, ester bonds are highly reactive, which makes them particularly susceptible to loss[2,3]. This reactivity can hamper their detection, raising the question of how many ester-linked PTMs remain

undiscovered or understudied. Unconventional ubiquitination that involves ester bonds with proteins, sugars and lipids[2,3] has recently emerged as a prime example of this intriguing family of PTMs.

Another interesting example is ADP-ribosylation (ADPr), a widespread and versatile PTM that regulates a variety of vital biological processes across all kingdoms of life[4]. Poly(ADP-ribose) Polymerases (PARPs) covalently attach ADP-ribose to various target amino acid side chains, leading to the formation of diverse chemical signals, including O-glycosidic and ester linked-ADPr[5]. This historically challenging PTM is mainly studied as the outcome of PARP1 signaling in the context of the DNA damage response (DDR)[6,7]. PARP1 detects DNA breaks and, in complex with HPF1[8,9], synthesizes mono- and poly-ADPr on serine residues of several target proteins, including histones and PARP1

[1]Research Group of Proteomics and ADP-Ribosylation Signaling, Max Planck Institute for Biology of Ageing, Joseph-Stelzmann-Strasse 9b, Cologne 50931, Germany. [2]Cologne Excellence Cluster for Stress Responses in Ageing-Associated Diseases (CECAD), University of Cologne, 50931 Cologne, Germany. [3]Present address: Department of Chemistry, Princeton University, Princeton, NJ, USA. ✉e-mail: Edoardo.Longarini@age.mpg.de; imatic@age.mpg.de

itself[10,11]. In addition, PARP1 alone, as well as virtually all the other members of the PARP family, can attach ADP-ribose to aspartate and glutamate residues, which have been historically seen as the primary ADPr targets for histones and PARP1[12]. The reversal of mono- and poly-ADPr is carried out by several known hydrolases[4,6]. In this context, ARH3 is known to hydrolyse the O-glycosidic linkage of serine ADPr (Ser-ADPr) and, to a lesser extent, poly-ADPr. By contrast, PARG degrades poly-ADPr and, according to the current consensus, is unable to remove the initial protein-ribose bond, leaving a mono-ADPr remnant on its targets[6,13–19]. However, this view has been recently challenged by evidence that recombinant PARG can cleave mono-ADPr from peptides[20]. This is supported by an earlier finding that PARG can completely degrade poly-ADPr, including the initial ADP-ribose moiety[21].

Over the last decade, a surge of methodologies and tools by various laboratories has significantly improved ADPr studies[22]. In particular, our laboratory has contributed by developing a phospho-guided chemoenzymatic strategy, resulting in the generation of a panel of site-specific and broad-specificity antibodies against mono-ADPr[14,17,23,24]. We have successfully applied these tools to provide important biological insights about serine ADPr[14,17]. Nonetheless, in spite of these technological advances, detection of more labile forms of ADPr, such as ester-linked ADPr, has remained difficult.

Here, we address this lack of adequate detection workflows for this family of labile PTMs. Focusing on Asp/Glu-ADPr, we establish a methodology that preserves it and enables its detection. By optimizing the sample processing steps for both immunoblotting and proteomics, we aimed to find practical solutions to address the lability of ester bonds. In our search for easy-to-implement conditions, we decided to adjust the temperature for cell lysis, as well as the pH and sample digestion time for proteomics. Specifically, for western blotting, to prevent the loss of Asp/Glu-ADPr, we adhered to the principle of never heating samples above room temperature, primarily keeping them at 4 °C. As a crucial aspect of our method, cell lysis was carried out at room temperature, avoiding the conventional boiling step, while still ensuring effective denaturing cell lysis that inactivates post-lysis activity of PARP1 and PARG. For proteomics analyses, we proposed short, acidic protein digestion at 37 °C enabled by the Arg-C Ultra protease, a protocol that effectively preserves ester-linked ADPr. Importantly, while previously identified Asp/Glu-ADPr sites are largely considered as poly-ADPr, typically made undistinguishable from mono-ADPr due to treatments with hydroxylamine, recombinant PARG or PDE[25–28], our proteomic approach specifically and unambiguously detects sites of mono-ADPr.

These advancements broaden the reach of our SpyTag modular antibodies[14,17] and substantially enhances the mapping of ester-linked modification sites by mass spectrometry, letting us observe mono-Asp/Glu-ADPr upon DNA damage and demonstrate its reversal by PARG. Importantly, this methodology can also be applied to the study of other PTMs containing ester bonds, as shown here for ester-linked ubiquitination.

## Results

### The high chemical stability of serine ADP-ribosylation
Historically, glutamate and aspartate were considered the primary acceptors for poly-ADPr in PARP1 signaling[5,7,12,27]. Our initial discovery of Ser-ADPr on histones[11] marked a significant shift in this perspective. Subsequent research from our team and others has established Ser-ADPr by the PARP1/HPF1 complex[8,10] as not only prevalent but also functionally important PTM[17,26,29–31]. This has inspired a surge of studies culminating in the notion that Ser-ADPr might be the only relevant signal of PARP1/2 upon DNA damage[25,26,29,32,33]. Consistently, in our own studies we were unable to detect mono-Asp/Glu-ADPr by western blotting[14,17]. Yet we were intrigued by a small but significant mono-ADPr signal detected by immunofluorescence in DNA damage-treated

HPF1 KO cells[17] and by a recent report on the lability of in vitro produced Asp/Glu-ADP-ribosylated peptides[20]. This led us to wonder if the absence of a detectable mono-Asp/Glu-ADPr signal might result not from absence of this modification in cells, but rather from the high chemical lability of its ester linkage compared to the O-glycosidic bond of Ser-ADPr (Fig. 1a).

Although we and others have effectively detected Ser-ADPr under acidic conditions[11,14,25,26,33,34], suggesting its stability, an early study showed that 44% formic acid rapidly releases ADP-ribose from a model conjugate, where ADP-ribose is attached to a free serine that is not embedded in a peptide or protein[35]. To reconcile this discrepancy and to exclude the possibility that Ser-ADPr is affected differently by various types of acid treatment, we replicated the exact conditions employed by Cervantes-Lauren et al.[35], by treating the widely used H3S10ADPr peptide[10,14] with 44% formic acid at 37 °C for 1 h. Under these conditions we observed by mass spectrometry no cleavage of O-glycosidic bonds from serines, while this linkage was efficiently cleaved by the Ser-ADPr hydrolase ARH3[36,37] (Fig. 1B, C, Supplementary Fig. 1 and Supplementary Data 1), as expected. The resistance of Ser-ADPr to highly acidic conditions was confirmed for two additional peptides and full-length histone H3 (Fig. 1D, Supplementary Fig. 1, and Supplementary Data 1). This implies that acid-induced loss of ADP-ribose from tissues observed by Cervantes-Lauren et al.[35] cannot be considered as evidence of Ser-ADPr. The higher-than-previously-thought chemical stability of Ser-ADPr under conditions commonly used in various sample preparation protocols suggests that Ser-ADPr could mask the detection of more labile forms of ADPr. This also may explain the success of Ser-ADPr peptides as a robust 'foundational technology' in the broad generation of ADPr antibodies[14,17,23].

### Preservation of ester-linked modifications reveals DNA damage-induced mono-Asp/Glu-ADPr
Having established the chemical stability of Ser-ADPr, even under conditions previously shown to hydrolyze O-glycosidic ADPr[35], we turned our attention to Asp/Glu-ADPr. While both the affinity-matured AbD43647 antibody and its parental clone AbD33204 were generated using Ser-ADPr peptides, they can also detect other types of mono-ADPr, including Asp/Glu-ADPr produced by recombinant PARP1[14,17]. We reasoned that the combination of a protocol preserving Asp/Glu-ADPr with the sensitive HRP-coupled AbD43647 format could effectively detect mono-Asp/Glu-ADPr in cells. Therefore, we investigated sample preparation conditions likely to affect the lability of Asp/Glu-ADPr—specifically pH, temperature and DNA shearing. We used cells lacking the Ser-ADPr cofactor HPF1[10] to avoid interference from the abundant, chemically stable Ser-ADPr. In standard sample preparation, samples are exposed to high temperatures to ensure protein denaturation. We posited that performing cell lysis with high concentrations of the denaturing agent SDS could eliminate the need to heat samples above room temperature. In HPF1KO cells, treated with $H_2O_2$ to activate PARP1, cell lysis with 4% SDS at room temperature—without boiling—markedly boosted the mono-ADPr signal (Fig. 2A). In contrast, in WT cells the mono-ADPr signal was mostly unaffected by boiling, reflecting the stability of Ser-ADPr at high temperatures, as noticed previously[14,17,20]. Importantly, protein extraction and immunoblotting efficiency, assessed via ponceau, PARP1, GAPDH and H3 staining, was not affected by the omission of boiling. Non-denaturing lysis conditions can result in the post-lysis activity of enzymes, including PARP1[11,38], requiring the use of inhibitors. Crucially, 4% SDS at room temperature is sufficient to completely inactivate PARP1 and PARG (Supplementary Fig. 2a). Therefore, our cell lysis protocol does not require the use of inhibitors (Supplementary Fig. 2b).

Encouraged by these results, we then evaluated methods to shear DNA and reduce the sample viscosity. We observed that for HPF1KO cells sonication, commonly used for shearing DNA, led to noticeable loss of mono-ADPr signal. This loss was mitigated by fragmenting DNA

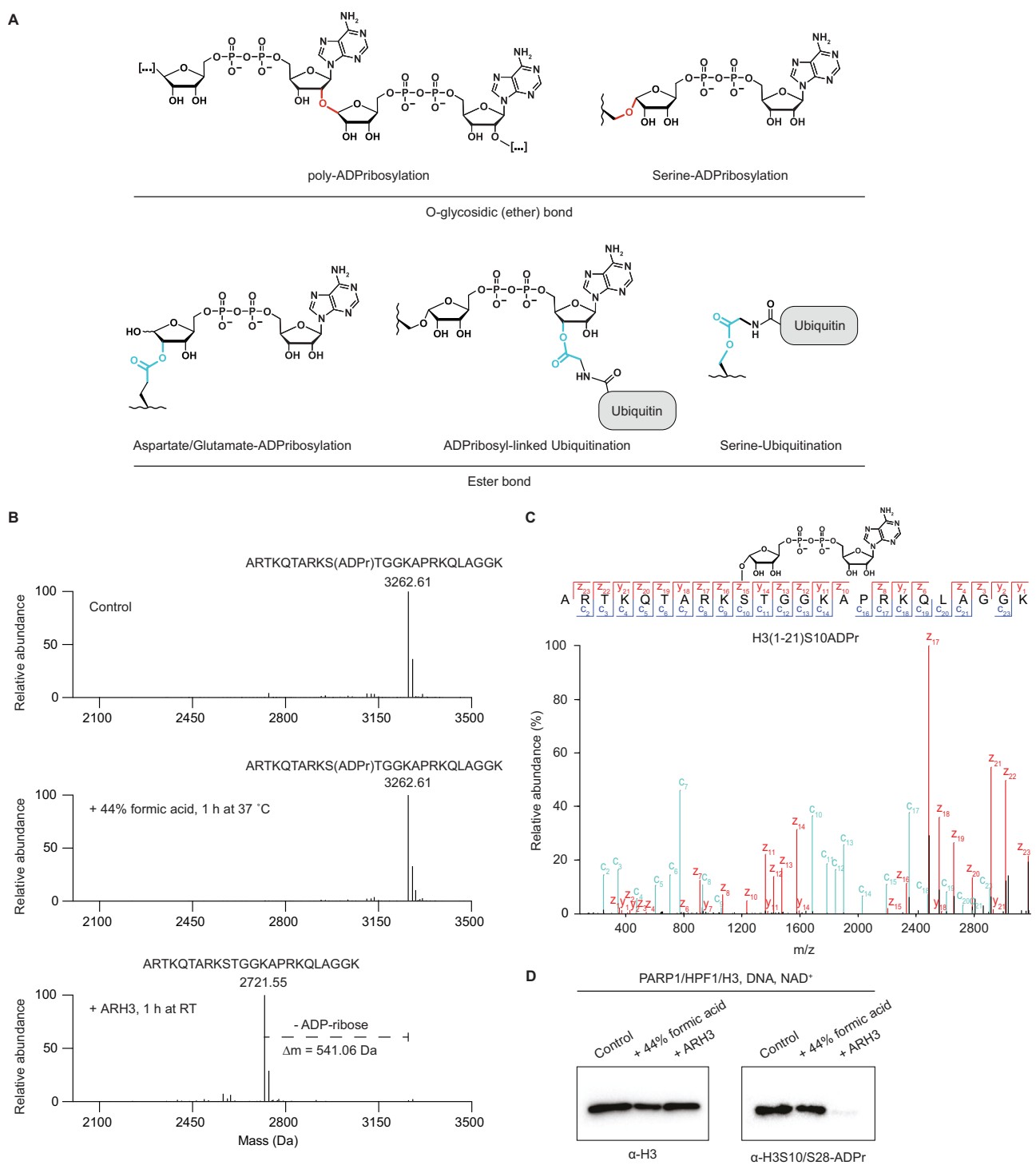

**Fig. 1 | The high chemical stability of serine ADP-ribosylation. A** Schematic illustration of various PTMs grouped by linkage types. Top: poly-ADP-ribose and Serine-ADP-ribose share a common O-glycosidic (ether) linkage, highlighted in red. Bottom: Asp/Glu-ADPr, ADP-ribosyl-ubiquitination and serine-ubiquitination share a common ester bond, highlighted in cyan.
**B** Deconvoluted MS spectrum of biotinylated Histone H3 (1-21) S10ADPr peptide untreated (top), treated with 44 % formic acid in water for 1 h at 37 °C (middle) or treated with recombinant ARH3 for 1 h at RT (bottom). As depicted, the in vitro ARH3 reaction produces a mass shift of 541.06 Da, corresponding to the loss of ADPr. Conversely, no appreciable loss of ADPr is observed in the

formic acid treated sample. See also Supplementary Fig. 1. **C** High-resolution Electron Transfer Dissociation (ETD) fragmentation spectrum of H3(1-21) S10ADPr peptide subjected to formic acid treatment, demonstrating localization of ADPribose to the H3S10 residue. **D** Immunoblotting of full length histone H3 ADPribosylated with PARP1:HPF1, stagetipped and dried to remove all buffer components, then treated with 44 % formic acid in water for 1 h at 37 °C, treated with recombinant ARH3 for 1 h at RT, or left untreated. The sample was stage-tipped a second time and dried before resuspension in loading buffer and immunoblotted with the indicated antibodies. Shown is a representative result from two independent experiments.

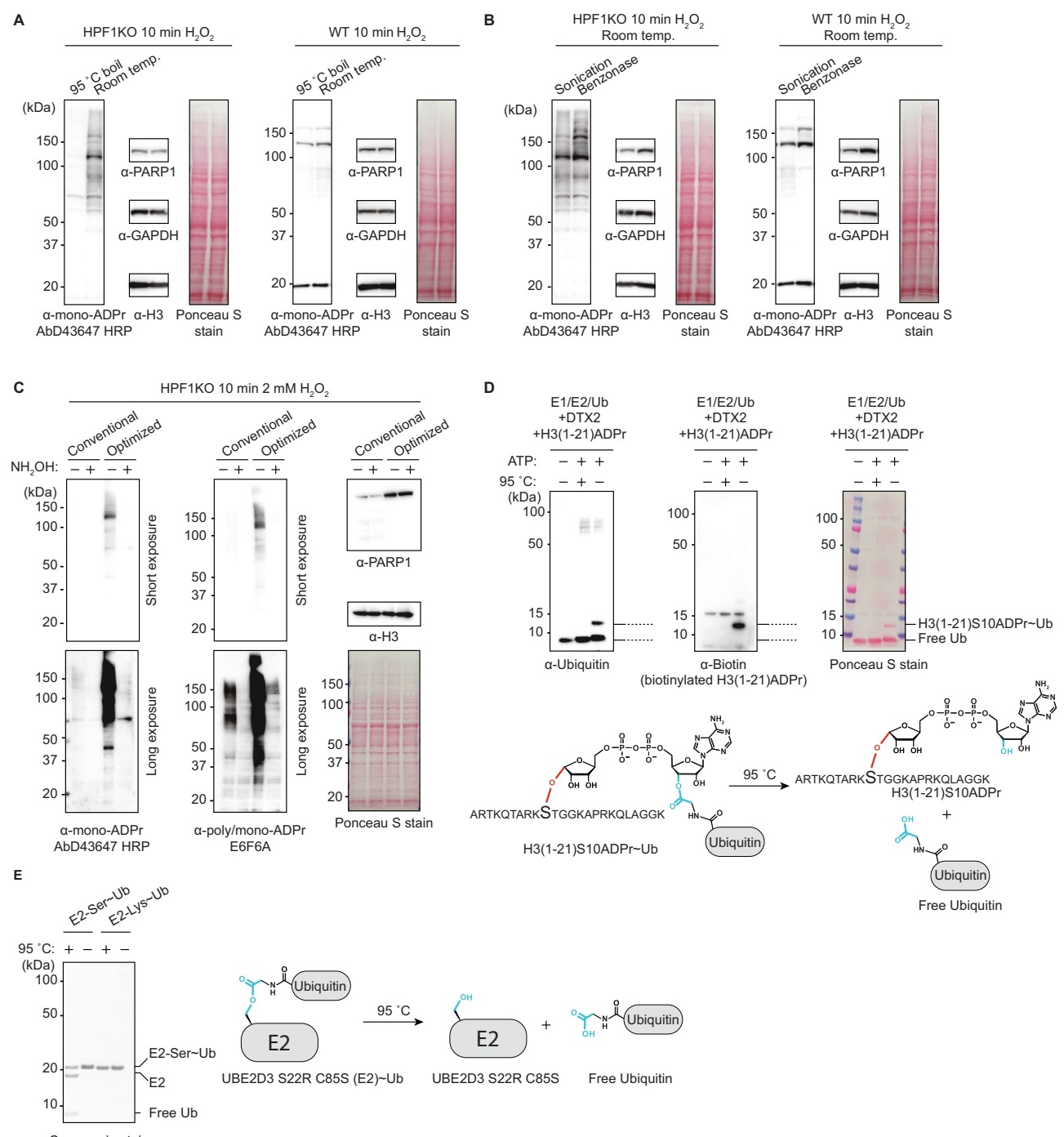

with recombinant benzonase instead of sonication (Fig. 2B). Interestingly, we found that sonication led to a reduction in total PARP1 levels, a phenomenon that was not observed with other proteins we tested. Lastly, we evaluated the effect of pH on Asp/Glu-ADPr stability. The standard pH of the lysis buffer, ranging from 7.2 to 7.9, appeared to have a negligible impact on the mono-ADPr signal, at least when the total sample processing time was kept under 1 h (Supplementary Fig. 2c). Additionally, while long room temperature incubations (≥2.5 h) at pH ~ 8.3 lead to substantial loss of Asp/Glu-ADPr, this loss is minimized with short (<30 min) incubation times (Supplementary Fig. 2d). This indicates that the lability of Asp/Glu-ADPr is influenced by a combination of factors—specifically, time, pH, and temperature—a conclusion that is corroborated by a recent study[20].

Overall, our optimized sample processing significantly improved detection of mono-ADPr in HPF1KO cell lysates (Fig. 2C and Supplementary Fig. 2e), a signal that was completely eliminated by treatment with hydroxylamine, which specifically cleaves Asp/Glu-ADPr but not Ser-ADPr[31]. Previously, to enhance antibody sensitivity for mono-ADPr, we used the SpyTag/SpyCatcher system to direct HRP conjugation to primary antibodies[17]. Here, we found that HRP-coupled AbD43647 vastly improved detection of mono-Asp/Glu-ADPr compared to its IgG counterpart, despite a tenfold lower concentration (Supplementary Fig. 2f). Lastly, we observed that prolonged incubation (3 h) at room temperature, a standard step during incubation of the secondary antibody, led to a slight reduction in the signal (Supplementary Fig. 2g). Consequently, skipping this step by using HRP-conjugated

**Fig. 2 | Preservation of ester-linked modifications reveals DNA damage-induced mono-Asp/Glu-ADPr. A** Immunoblotting images showing the effect of boiling during sample processing on mono-ADPr signal. HPF1KO U2OS cells were treated with 2 mM $H_2O_2$ for 10 min to induce DNA damage, following by harvesting and lysis according to standard procedures, then either boiled at 95 °C for 10 min or left at room temperature. Shown is a representative result from four independent experiments. **B** Immunoblotting images showing the effect of DNA fragmentation by sonication or benzonase on mono-ADPr signal. HPF1KO U2OS cells were treated and harvested as in **2a** without boiling, then either sonicated at 4 °C or treated with recombinant benzonase at RT for 10 min. Shown is a representative result from four independent experiments. **C** Immunoblotting images comparing overall cell lysis procedures. HPF1KO U2OS cells were $H_2O_2$-treated and harvested according to conventional procedures (sonication, pH 8.0, 95 °C boil) or with our optimized process (benzonase, pH 7.2, room temp.). Where indicated, the samples were treated with 1 M $NH_2OH$ (hydroxylamine) for 2 h at room temp. to remove Asp/Glu-ADPr. See also Supplementary Fig. 2e. Shown a representative result from three independent experiments. **D** Immunoblotting images showing hydrolysis of a hybrid ADP-ribosyl-ubiquitin peptide construct during routine immunoblotting conditions. Top: The ADP-ribosylated peptide (H3(1-21)S10ADPr) conjugated to the C-term of ubiquitin via the 3' hydroxyl group of the adenine-proximal ribose is incubated for 10 min at 95 °C or at room temperature and immunoblotted. Bottom: schematic illustration of the H3S10ADPr-Ub construct and corresponding hydrolysis products. Shown a representative result from two independent experiments. **E** SDS-PAGE showing hydrolysis of serine-linked ubiquitination during routine immunoblotting conditions. Right: recombinant UBE2D3 S22R C85S (E2) is ester-linked to the C-terminus of ubiquitin (Ub) with the C85S residue. UBE2D3 S22R C85K linked to the C-term of Ub with a canonical isopeptide bond was used as a control. Each E2-Ub construct was incubated for 10 minutes at 95 °C or at room temperature, run on SDS-PAGE gel and visualized with Coomassie stain. Left: schematic illustration of the E2-Ser-Ub construct and corresponding hydrolysis. Shown a representative result from two independent experiments. For (**A**, **B**) ponceau S staining shows total protein load.

primary antibodies in our modular format[17] represents an additional advantage.

The ester linkage that characterizes Asp/Glu-ADPr is also found in many other emerging PTMs, such as ubiquitination of serine residues[2,3] and ADP-ribose-linked ubiquitination[39] (Fig. 1A). We reasoned that if the chemical lability is a general property of this family of PTMs, our optimized protocol would also enhance the detection of other ester-linked modifications. To explore this possibility, we used the H3S10ADPr peptide[10,14] to prepare the hybrid, ester-linked ADP-ribosyl-ubiquitin modification (Fig. 1A) as previously described[39]. Exposure of the resulting ADP-ribose-linked ubiquitinated H3S10ADPr peptide to boiling caused a dramatic loss of ubiquitination, which was prevented by skipping the boiling step (Fig. 2D). Next, we explored ester-linked serine ubiquitination using an established model system in which ubiquitin is attached to the serine of a modified recombinant Ub-conjugating enzyme (E2-Ser-Ub)[40,41]. Consistent with our hypothesis, boiling of E2-Ser-Ub resulted in the loss of ubiquitin, a phenomenon which was not observed with canonical isopeptide-linked ubiquitination (Fig. 2E).

Overall, we have developed a protocol that effectively preserves the highly labile mono-Asp/Glu-ADPr, enabling the detection of this elusive PTM. Enabled by this method, we have demonstrated that this modification is prevalent in HPF1KO cells, where Ser-ADPr is dramatically reduced[10]. Our findings provide the first conclusive cellular evidence that HPF1 switches the amino acid target specificity of PARP1 from Asp/Glu- to Ser-ADPr, a concept derived from biochemical assays[10,42]. As an illustration of the significance of our approach in exploring biology, the previous inability to preserve and, therefore, detect Asp/Glu-ADPr led to the puzzling conclusion that HPF1 does not switch the amino acid target specificity of PARP1 from Asp/Glu- to Ser-ADPr[33].

In broad terms, we show that loss under routine sample processing conditions is a general property of ester-linked PTMs and demonstrate that our modified sample processing can greatly enhance their detection.

## Short acidic protein digestion with Arg-C Ultra preserves mono-Asp/Glu-ADPr for proteomic analyses

Building upon the methodology described above for the successful detection of the mono-Asp/Glu-ADPr using our SpyTag modular antibodies (Fig. 2), we concluded that standard proteomic workflows, which involve extended protease incubation at 37 °C under mildly alkaline pH conditions, are not suitable for investigating ester-linked PTMs. To test this, we exposed the recombinant E2-Ser-Ub construct to a range of pH and temperature and observed loss of ester-linked ubiquitin, but not of canonical isopeptide-linked ubiquitin, under overnight incubation at 37 °C and mild alkaline pH (pH 8.0) (Supplementary Fig. 3a). By contrast, the loss of ubiquitin was minimized by short incubation at mild acidic conditions (pH 5.1). Given these results, we reasoned that preserving mono-Asp/Glu-ADPr for proteomic analyses could be achieved by implementing an approach involving short acidic protein digestion with Arg-C Ultra (Promega). This recently introduced arginine-specific protease significantly outperforms the conventional Arg-C in terms of specificity and efficiency, making it an important addition alongside trypsin and Lys-C to the limited number of proteases widely applicable for various proteomics applications. An important advantage of Arg-C Ultra compared to trypsin in the context of ester-linked modifications is its excellent efficiency with short digestion times and low pH. To obtain trypsin-like specificity we combined Arg-C Ultra with Lys-C (Fig. 3A). Benchmarked on automodified recombinant PARP1E988Q, this method yields sequence coverage comparable to standard trypsin digestion and a lower number of misscleavages (Supplementary Fig. 3b, c). Importantly, this workflow results in a ~10-fold improvement in mono-Asp/Glu-ADPr detection (Fig. 3B, Supplementary Data 1), allowing us to identify >40 high-confidence Asp/Glu-ADPr sites, more than any previously reported with the intact ADPr moiety (Supplementary Fig. 3d, Supplementary Data 1). With our optimized proteomics workflow at hand, we sought to directly identify Asp/Glu-ADPr sites in cells. The detection of a prominent band between 100 and 150 kDa (Fig. 2), corresponding to the observed migration of mono-ADP-ribosylated PARP1, suggests that PARP1 might be one of the most abundant mono-Asp/Glu-ADPr substrates. Therefore, we immunoprecipitated GFP-tagged PARP1 from $H_2O_2$-treated HPF1KO cells and processed the resulting eluate with our optimized proteomics method. We identified 10 high-confidence mono-Asp/Glu-ADPr sites on PARP1 (Fig. 3C, D and Supplementary Data 1), unequivocally demonstrating mono-Asp/Glu-ADPr formation upon DNA damage. This represents a significant improvement over a recent large-scale proteomics study which did not identify any PARP1 Asp/Glu-ADPr after ADPr enrichment from HPF1KO cells[33]. Our results affirm the validity of previous approaches that have identified, in cellular contexts and with recombinant PARP1, many of the major Asp/Glu-ADPr sites reported here[27,28,43–47]. Intriguingly, mutating all target glutamates present in the automodification domain did not result in a noticeable decrease in total PARP1 mono-ADPr (Supplementary Fig. 3e). Mass spectrometry revealed a compensatory increase at other sites (Supplementary Fig. 3f and Supplementary Data 1), reminiscent of ubiquitination, which often occurs at alternative lysines when target lysines are mutated[48,49].

Notably, we observed that HCD fragmentation of Asp/Glu-ADPr peptides causes preferential breakage of the diphosphate group of ADP-ribose, rather than the complete loss of the ADP-ribose moiety. This leads to the loss of AMP and retention of a phosphoribose-$H_2O$ remnant on the modified amino-acid (Fig. 3D). In practice, this facilitates the localization of Asp/Glu-ADPr sites with high-quality HCD fragmentation data. This behavior is in contrast to Ser-ADPr peptides, where HCD fragmentation leads largely to complete loss of the modifier, leaving no localization-specific ions[50].

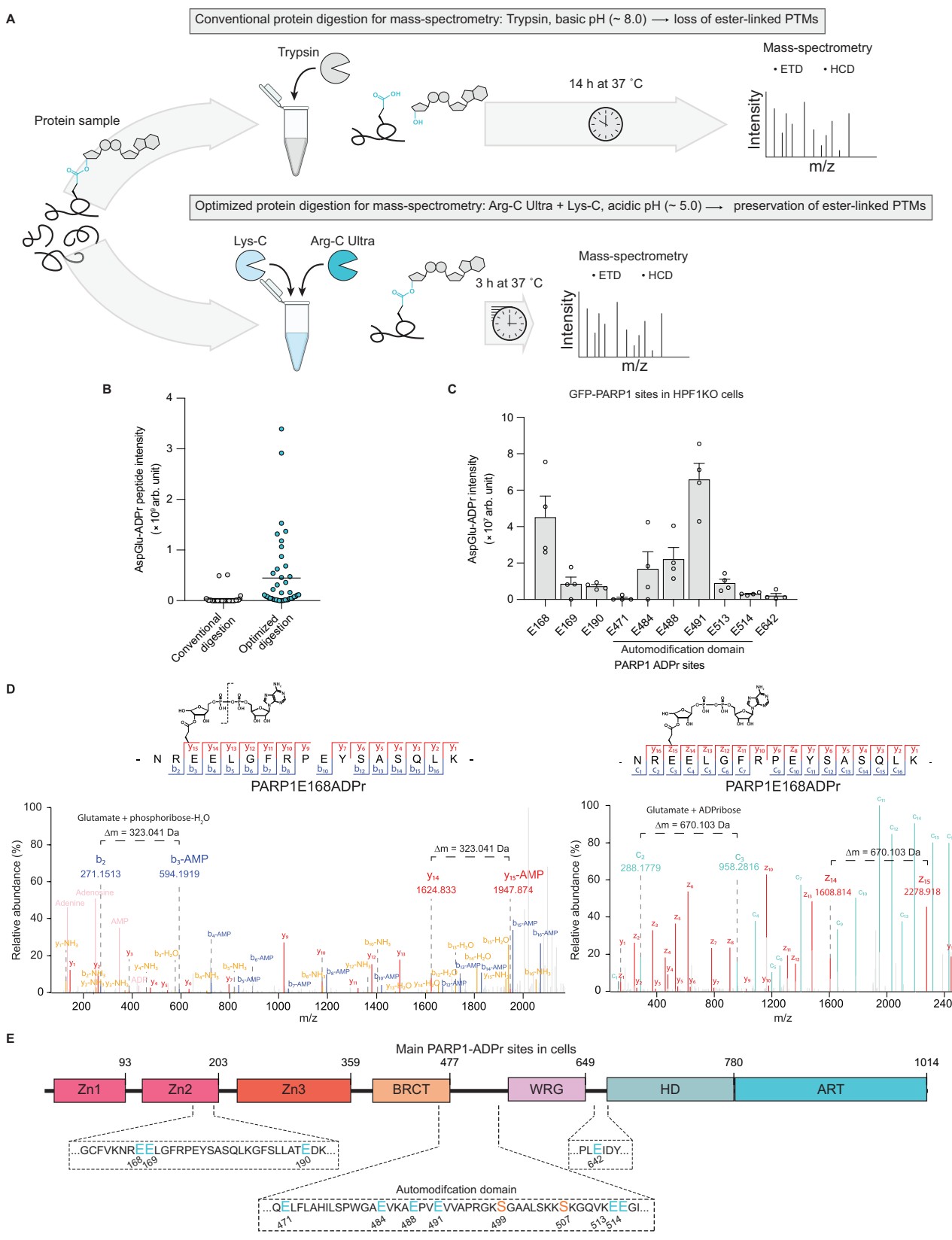

**A** Conventional protein digestion for mass-spectrometry: Trypsin, basic pH (~ 8.0) ⟶ loss of ester-linked PTMs

Optimized protein digestion for mass-spectrometry: Arg-C Ultra + Lys-C, acidic pH (~ 5.0) ⟶ preservation of ester-linked PTMs

**B**

**C** GFP-PARP1 sites in HPF1KO cells

**D** PARP1E168ADPr — Glutamate + phosphoribose-$H_2O$ Δm = 323.041 Da

PARP1E168ADPr — Glutamate + ADPribose Δm = 670.103 Da

**E** Main PARP1-ADPr sites in cells

As expected, given the high stability of Ser-ADPr in acidic conditions (Fig. 1B–D), the acid digestion approach also works for this type of ADPr. Specifically, mass spectrometric analysis of GFP-PARP1 immunoprecipitated from WT cells enabled the simultaneous detection of both Ser- and Asp/Glu-ADPr sites (Fig. 3E, Supplementary Fig. 3g).

## Identification of mono-Asp/Glu-ADPr on additional targets through ester bond preservation

Next, we sought to extend the identification of mono-Asp/Glu-ADPr sites to additional protein targets beyond PARP1. Towards this aim, we set out to combine the acidic digestion protocol with peptide immunoprecipitation based on the IgG format of AbD43647 (Fig. 4A), the

**Fig. 3 | Short acidic protein digestion with Arg-C Ultra preserves mono-Asp/Glu-ADPr for proteomic analyses. A** Schematic overview of standard mass-spectrometry sample preparation workflow (top) or of our updated workflow for the preservation of ester-linked PTMs (bottom). A key feature of our protocol is the short (~3 h) incubation at acidic pH with the proteases Arg-C Ultra and Lys-C, which provide the optimal environment for preservation of ester-linked PTMs while ensuring high digestion efficiency. This schematic was generated in Adobe Illustrator using some graphical elements adapted from Longarini et al.[17] **B** Intensity of Asp/Glu-ADPr peptides, derived from automodified PARP1E988Q, identified with the conventional "Trypsin" or our optimized "Arg-C Ultra + Lys-C" protocols (**A**). Data is expressed as dot-plot with each dot representing the corresponding intensity of all ($n = 56$) identified ADPr sites, and the bar representing sample mean, from a representative of 4 independent experiments. **c** Average abundance of high-confidence (delta score > 40, localization probability ≥ 0.9) mono-ADPr sites identified in GFP-PARP1 from HPF1KO cells. Data is expressed as mean ± SEM of $n = 4$ independent replicates. **D** Left: high-resolution High-energy Collisional Dissociation (HCD) fragmentation spectrum of a PARP1 ADPribosylated peptide identified from cells (**C**). HCD fragmentation of the Asp/Glu-ADPr peptide leads to loss of AMP ($\Delta$mass = −347.06 Da) and leaves phosphoribose-$H_2O$ ($\Delta$mass = 193.998) as a diagnostic fragment that can be used to identify the site of ADPribosylation. Left: corresponding high-resolution ETD spectrum of the same peptide species in which Asp/Glu-ADPr is preserved and can unambiguously assign the modification site. **E** Schematic depiction of all high-confidence (delta score > 40, localization probability ≥ 0.9) PARP1 Glu- and Ser-ADPr sites identified in cells displayed a linear model of PARP1 (Zn = zinc-finger domain, HD = autoinhibitory domain, ART = catalytic domain). Data from Fig. **C** and Supplementary Fig. 3g.

high-affinity mono-ADPr antibody not previously applied in proteomics[17]. While the cost of Arg-C Ultra and Lys-C significantly exceeds that of trypsin, limiting their use for digesting large amounts of proteins, early evidence suggests Arg-C Ultra's effectiveness at higher dilutions than typical for other proteases, offering significant potential for cost reductions. Our analysis showed that even with a very high protease-to-protein ratio of 1:2000, the increase in the number of miscleavages was marginal (Supplementary Fig. 4a), thereby enabling acidic digestion of milligram of proteins for ADPr enrichment (Fig. 4A). Given the sensitive detection of mono-ADPr by AbD43647[17] along with the successful application of its parental clone in proteomics[14], we chose its IgG format for immunoprecipitation of ADP-ribosylated peptides before mass spectrometry. Upon treating HPF1KO cells with $H_2O_2$, we identified 203 high-confidence sites (localization probability ≥ 90%) on 86 protein targets, with 151 sites on Asp/Glu residues (Fig. 4B–H, Supplementary Figs. 4, 5 and Supplementary Data 2). Compared to existing large-scale proteomics datasets[25–28,33,46,51], at the modification site level we revealed 114 novel sites (Fig. 4B, E–H and Supplementary Data 2). Some of the identified sites reside on lysine-rich peptides that would be challenging to identify with trypsin, highlighting the complementary nature of using Arg-C Ultra only (Fig. 4F, H). The mono-Asp/Glu-ADPr sites mapped to 69 protein targets, the majority of which were previously identified in the context of Ser-ADPr, indicating the homogeneous nature of PARP1 substrate targeting. We also observed histone Asp/Glu-ADPr (Fig. 4D, Supplementary Fig. 5a and Supplementary Data 2), confirming that Asp/Glu-ADPr can also occur on the nucleosomal surface as indicated by previous reports[52–56]. Additionally, we identified 18 previously unreported protein targets for Asp/Glu-ADPr (Supplementary Fig. 4c and Supplementary Data 2). Although Asp/Glu residues have traditionally been associated almost exclusively with poly-ADPr, our mono-ADPr proteomics analysis has revealed that these identified sites can also serve as targets for monomeric ADPr.

## PARP1 is the writer of DNA damage dependent mono-Asp/Glu-ADPr
Using our refined detection method, we explored whether PARP1 is the writer of mono-Asp/Glu-ADPr upon DNA damage. This is supported by our detection of PARP1 automodification sites in HPF1KO cells (Figs. 3 and 4). As the predominant ADP-ribosyl transferase in the DDR, PARP1 accounts for most mono- and poly- Ser-ADPr. Yet, other enzymes, including PARP2, PARP3, PARP10, and PARP14, may also contribute to mono-Asp/Glu-ADPr formation in the DDR. Thus, we aimed to determine the extent of PARP1's contribution to this signal. Treating HPF1KO cells with Olaparib, a PARP1/2 inhibitor, resulted in the abolition of mono-Asp/Glu-ADPr (Fig. 5A). To distinguish PARP1 from PARP2, we used the PARP1-specific inhibitors AZD5305 and AZD9574[57,58]. As expected, in WT cells, PARP1 inhibition abolishes the mono-Ser-ADPr signal (Fig. 5B, C), consistent with our previous observations using PARP1KO and PARP1PARP2DKO[14]. In HPF1KO cells, PARP1-specific inhibitors completely abolished the mono-Asp/Glu-

ADPr signal (Fig. 5D, E), indicating PARP1 as the primary source of this modification upon DNA damage.

## Human PARG can hydrolyse mono-Asp/Glu-ADPr in cells
Having established the dependency of mono-Asp/Glu-ADPr on PARP1 (Fig. 5), we then focused on exploring the effect of the poly-ADP-ribose hydrolase PARG on this PTM. Utilizing our anti-mono-ADPr antibodies, which exhibit no cross-reactivity with poly-ADPr[17], we previously demonstrated that formation of mono-Ser-ADPr is partially dependent on the PARG-mediated conversion of poly- to mono-ADPr[14,17]. Therefore, we hypothesized that mono-Asp/Glu-ADPr could result from a similar mechanism, anticipating that inhibiting or depleting PARG would reduce mono-Asp/Glu-ADPr levels, mirroring the effects seen with mono-Ser-ADPr[14,17]. To verify this hypothesis, we exposed HPF1KO cells to a PARG inhibitor (PARGi). As anticipated, poly-ADPr increased in DNA damaged cells after PARGi treatment, with the total PARP1 signal smeared across the membrane due to its high poly-ADPr status. Unexpectedly, however, PARGi also sharply increased mono-Asp/Glu-ADPr levels (Fig. 6A). To directly compare the effects of PARG on both Ser and Asp/Glu mono-ADPr, we proceeded to test the impact of PARGi in wild-type (WT) cells. Aligning with our previous studies[14,17], PARGi treatment dramatically decreased mono-Ser-ADPr, especially on PARP1 and core histones, alongside a rise in poly-ADPr (Fig. 6B). However, similar to the findings in HPF1KO cells, treating WT cells with PARGi also increased mono-Asp/Glu-ADPr. Importantly, this indicates that the formation of mono-Asp/Glu-ADPr is not limited to HPF1 KO cells but also occurs in WT cells. In WT cells poly-ADPr is largely present on serine residues and therefore remains intact upon hydroxylamine treatment, as shown by a poly-ADPr antibody[59]. In contrast, hydroxylamine treatment substantially removed the signal detected by AbD43647, further validating the mono-ADPr specificity of this antibody and indicating that the signal observed in WT and HPF1KO cells is not due to cross-reactivity with poly-ADPr. Immunofluorescence confirmed PARGi increases mono-Asp/Glu-ADPr, but hydroxylamine only partially reduced this signal (Fig. 6C), indicating PARG may also regulate other types of ADPr, including DNA/RNA ADPr[60], not detectable by western blotting. We then used PARG knockouts (PARGKO) to validate that the increase in mono-Asp/Glu-ADPr we observed upon PARG inhibition is not due to off-target effects. In agreement with PARGi data, PARGKO cells show a similar increase in mono-Asp/Glu-ADPr and treatment with PARGi in PARGKO cells did not result in any appreciable increase in signal (Fig. 6D).

These observations indicate that PARG can hydrolyze mono-Asp/Glu-ADPr in cells, a surprising finding in light of several earlier studies which showed that in biochemical reactions PARG cleaves poly-ADPr but cannot remove the last ADP-ribose attached to protein targets[13–16,18,19]. To resolve the discrepancy between our findings in cells and previous results from biochemical assays, we conducted a biochemical characterization of PARG, encouraged by a recent report suggesting that recombinant PARG can remove mono-Asp/Glu-ADPr

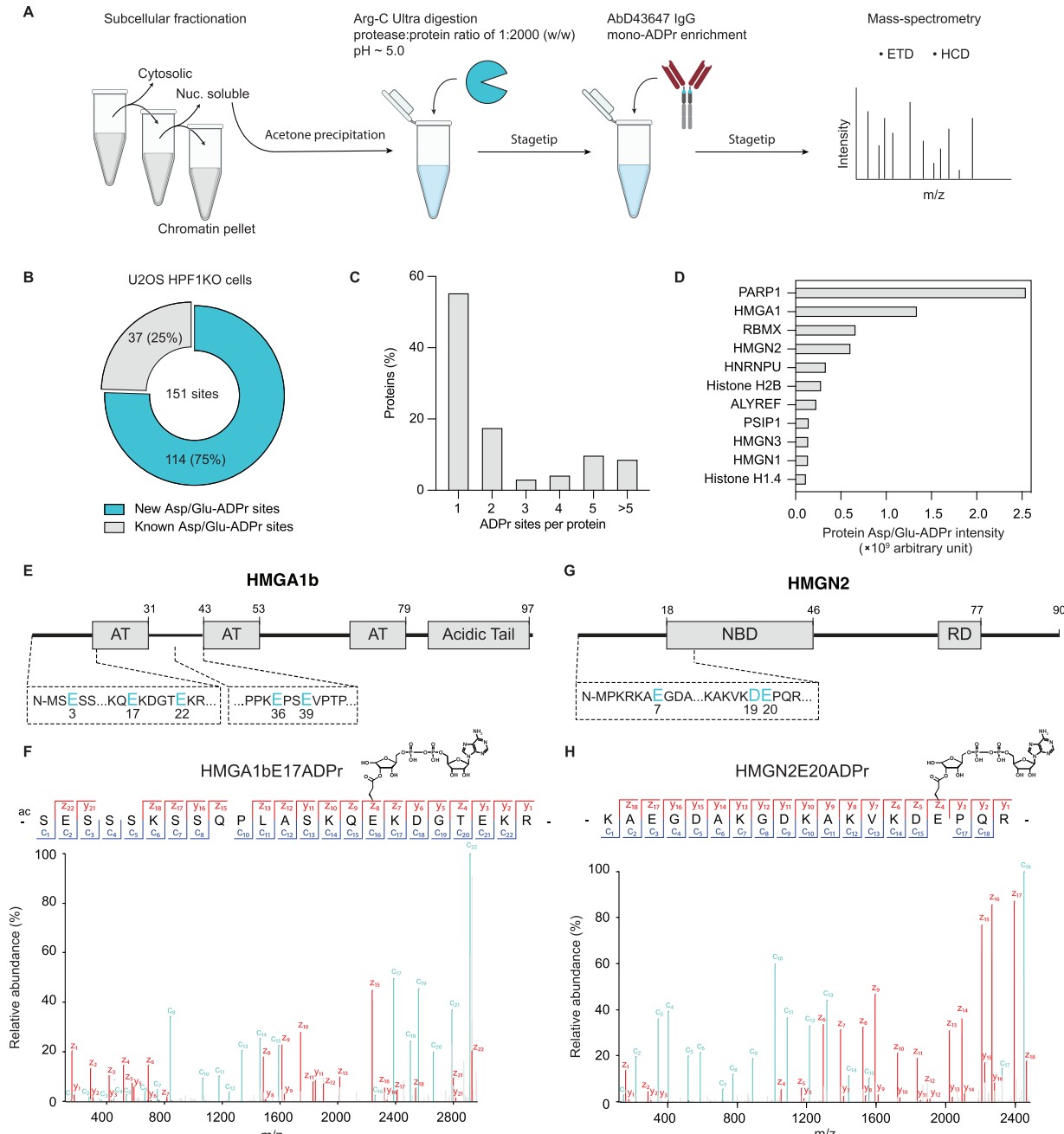

**Fig. 4 | Identification of mono-Asp/Glu-ADPr on additional targets through ester bond preservation. A** Schematic overview of the experimental workflow for enrichment of mono-ADPr peptides. Arg-C Ultra at a high protease-to-protein ratio (1:2000) is used for protein digestion, followed by peptide purification and mono-ADPr enrichment by the AbD43647 IgG antibody. Subsequent mass-spectrometry employs both HCD- and ETD-type fragmentation methods for high-confidence localization of the modification site. This schematic was generated in Adobe Illustrator using some graphical elements adapted from Longarini et al.[17] **B** Pie-chart overview of the localized Asp/Glu-ADPr sites (delta score > 40, localization probability ≥ 0.9) from mono-ADPr enrichment in HPF1KO cells and comparison with published studies (See also Supplementary Data 2). **C** Overview of the number of ADPr sites per protein. **D** Top 11 proteins identified in our large-scale mono-ADPr

enrichment experiments ranked by the cumulative intensity of the identified ADPr sites. **E** Schematic depiction of all high-confidence (delta score > 40, localization probability ≥ 0.9) HMGA1b mono-Asp/Glu-ADPr sites identified in HPF1KO cells on a linear model of HMGA1b. These sites have not been previously identified in other large-scale proteomic studies. **F** ETD spectrum showing unambiguous localization of the HMGA1bE17ADPr modification site. **G** Schematic depiction of all high-confidence (delta score > 40, localization probability ≥ 0.9) HMGN2 mono-Asp/Glu-ADPr sites identified in HPF1KO cells on a linear model of HMGN2 (NBD Nucleosome Binding Domain, RD Regulatory Domain). These sites have not been previously identified in other large-scale proteomic studies. **H** ETD spectrum showing unambiguous localization of the HMGN2E20ADPr modification site. Data for (**B**–**H**) is from a total of three biological replicates.

from in vitro modified peptides[20] and by the complete removal of PARP1 automodification observed in PARG zymograms[21]. First, we used automodified PARP1E988Q as a source of mono-Asp/Glu-ADPr. Upon subsequent addition of PARG, we observed a time- and concentration-dependent removal of mono-Asp/Glu-ADPr

(Supplementary Fig. 6a, b), an effect that is abolished in the presence of PARGi (Supplementary Fig. 6c). This effect becomes noticeable only after long incubation times, which might also explain why in earlier reports, using more conservative conditions, this phenomenon was not observed[13,18,19]. Next, we automodified PARP1 WT using

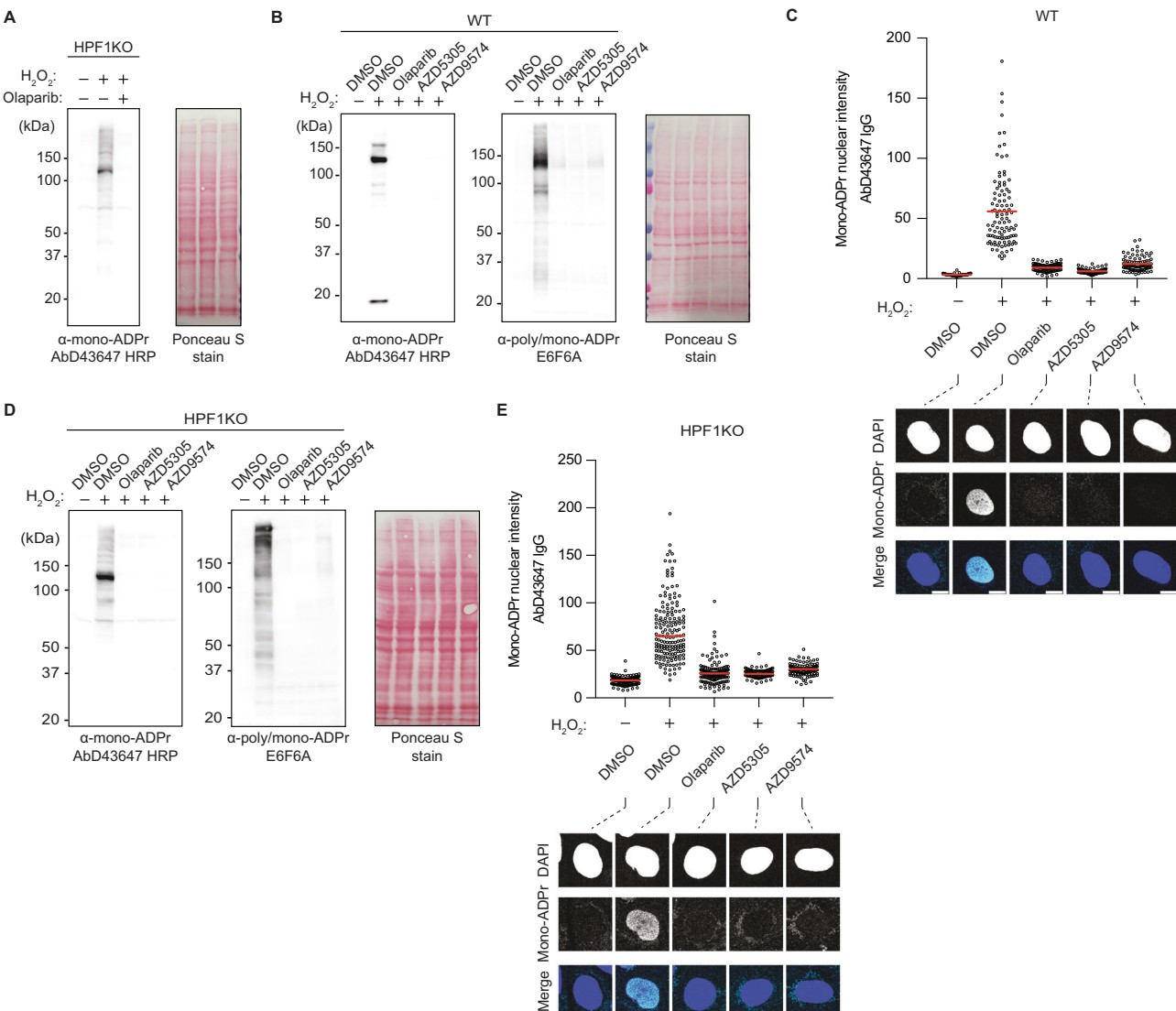

**Fig. 5 | PARP1 is the writer of DNA damage dependent mono-Asp/Glu-ADPr.**
**A** HPF1KO U2OS cells were treated with 1 µM Olaparib for 30 min to inhibit PARP1/2 or with DMSO control, followed by 2 mM $H_2O_2$ for 10 min as indicated to induce DNA damage, then harvested and lysed according to our optimized procedure, immunoblotted with the indicated antibodies and stained with Ponceau S to show total protein load. **B** WT U2OS cells were treated with 1 µM Olaparib for 30 min to inhibit PARP1/2, 10 nM AZD5303 or AZD9475 for 30 min to selectively inhibit PARP1, or with DMSO control, followed by 2 mM $H_2O_2$ for 10 min as indicated to induce DNA damage, then harvested and lysed according to our optimized procedure, immunoblotted with the indicated antibodies and stained with Ponceau S

to show total protein load. **C** Top: immunofluorescence quantification of WT U2OS cells treated as indicated, then fixed with methanol and stained with AbD43647 IgG and DAPI. The signal intensity of ~100 cells from a representative of 2 independent experiments is shown. The red bars indicate sample mean. Bottom: representative confocal images. Scale bar = 10 µm. **D** HPF1KO U2OS cells were processed and immunoblotted as indicated. **E** Top: immunofluorescence quantification of HPF1KO U2OS cells processed as indicated. The signal intensity of ~100 cells from a representative of 2 independent experiments is shown. The red bars indicate sample mean. Bottom: representative confocal images. Scale bar = 10 µm. For figures (**A**–**C**) shown is a representative result from three independent experiments.

conditions which generate poly-ADPr and no detectable mono-ADPr. Subsequent addition of PARG resulted, as expected, in rapid degradation of poly-ADPr to mono-ADPr. The rapid poly-ADPr hydrolysis is then followed by a slower and incomplete removal of mono-ADPr (Supplementary Fig. 6d), consistent with our previous results (Supplementary Fig. 6a). Using mass spectrometry, we observed that PARG almost completely removed mono-Asp/Glu-ADPr from peptides, without a clear preference for specific sites (Supplementary Fig. 6e, f). To characterize the activity of PARG more broadly, we investigated removal of cellular mono-Asp/Glu-ADPr by recombinant PARG. To this end, we utilized an immunofluorescence approach previously employed for quantifying the on-slide removal of ADPr by ARH3 and snake venom phosphodiesterase[17]. Complete removal of ADPr from HPF1 KO cells was achieved with much lower

concentration—down to as low as 62 nM—of recombinant PARG compared to the in vitro experiment (Fig. 6E).

These findings show PARG can hydrolyze the ester linkage of Asp/Glu-ADPr, although less efficiently than the O-glycosidic bond in poly-ADPr. However, this activity of PARG appears more significant in cells, as seen by the mono-Asp/Glu-ADPr increase upon PARG inhibition. Additionally, our findings reveal that mono-Asp/Glu-ADPr is constantly formed and hydrolyzed in WT cells, with PARG playing a key role in maintaining the balance of mono-ADPr on serine and Asp/Glu residues. It does so by simultaneously promoting the formation of mono-Ser-ADPr, through the degradation of poly-ADPr to mono-Ser-ADPr, and by facilitating the removal of mono-Asp/Glu-ADPr. Conversely, inactivating PARG suppresses mono-Ser-ADPr formation and shifts the balance towards mono-Asp/Glu-ADPr.

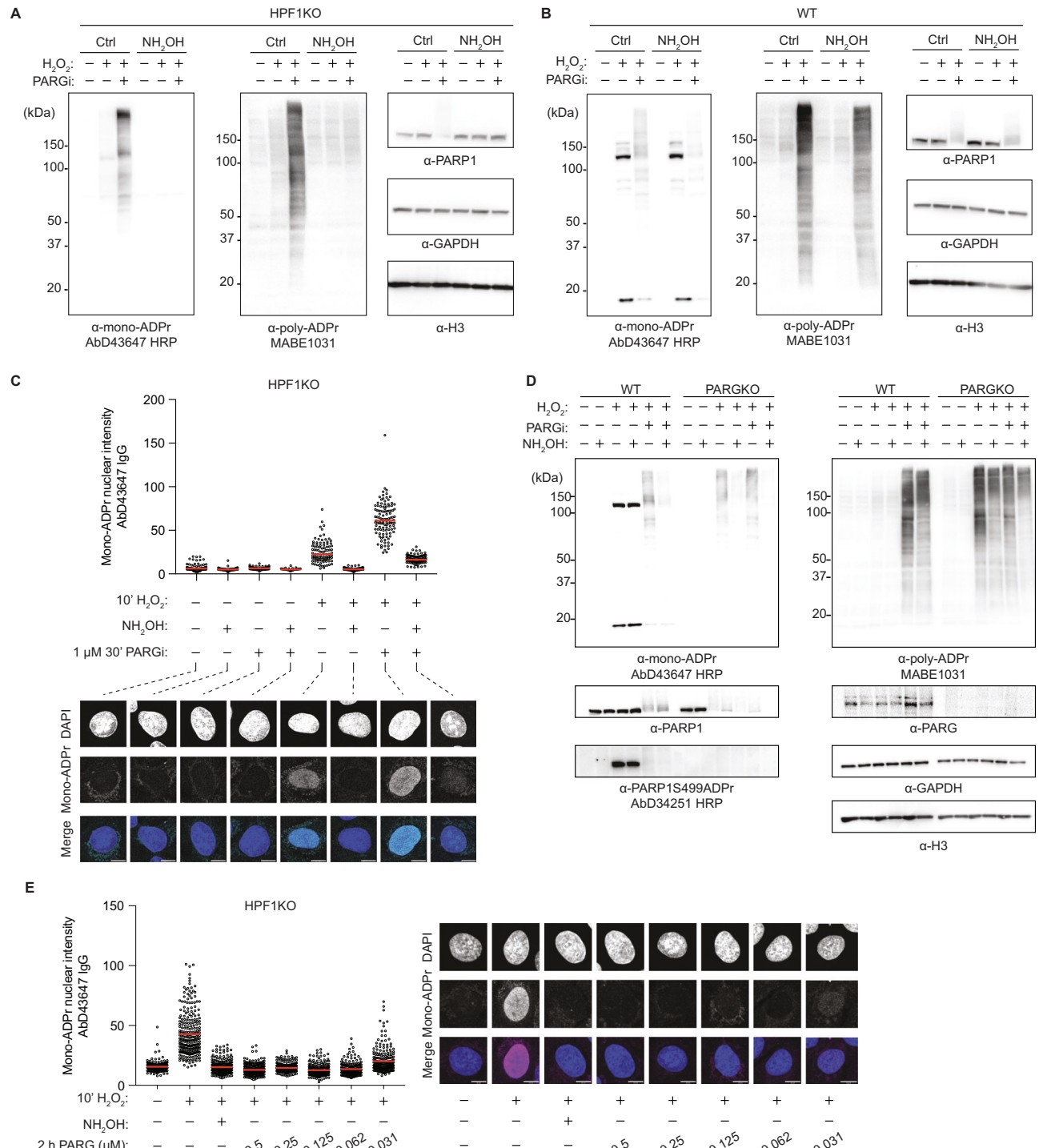

**Fig. 6 | Human PARG can hydrolyse mono-Asp/Glu-ADPr in cells. A** HPF1KO U2OS cells were treated with 1 μM PARGi (PDD00017273, Sigma) or with DMSO control for 30 min, followed by 2 mM H₂O₂ for 10 min as indicated to induce DNA damage, then harvested and lysed according to our optimized procedure. The samples were treated, where indicated, with 1 M NH₂OH for 2 h at room temp. and immunoblotted with the indicated antibodies. Shown is a representative result from three independent experiments. **B** WT U2OS cells were processed and immunoblotted as in (**A**). Shown is a representative of result from three independent experiments. **C** Top: Immunofluorescence quantification of HPF1KO U2OS cells treated as indicated, fixed with methanol and treated with or without 1 M NH₂OH for 2 h at room temp. before blocking and staining with the indicated antibody and DAPI. The signal intensity of ~100 cells from a representative of 3

independent experiments is shown. Red bars indicate sample mean. Bottom: representative confocal images. Scale bar = 10 μm. **D** WT or PARGKO U2OS cells were treated with 1 μM PARGi (PDD00017273, Sigma) or with DMSO control for 30 min, then with 2 mM H₂O₂ for 10 min as indicated to induce DNA damage, followed by harvesting and lysis according to our optimized procedure. The samples were treated, where indicated, with 1 M NH₂OH for 2 h at room temp. and immunoblotted with the indicated antibodies. Shown is a representative result from three independent experiments. **E** Immunoblotting images showing removal of mono-Asp/Glu-ADPr signal upon on-slide recombinant PARG treatment of H₂O₂-treated HPF1KO cells. Right: the signal intensity of ~200 cells from a representative of 3 independent experiments is shown. Red bars indicate sample mean. Left: representative confocal images. Scale bar = 10 μm.

### Mono-Asp/Glu-ADPr belongs to the initial wave of PARP1 signaling

Having established PARG as an eraser of mono-Asp/Glu-ADPr, we sought to investigate the DNA damage-dependent Asp/Glu-ADPr dynamics. Therefore, we treated cells with $H_2O_2$ for different durations and prepared the samples with our Asp/Glu-ADPr-preserving protocol. Considering that AbD43647 can recognize both Ser- and Asp/Glu-mono-ADPr, we used the PARP1S499ADPr (AbD34251)[14] antibody, which does not recognize Asp/Glu-ADPr, to distinguish and compare the dynamics of Asp/Glu-ADPr and Ser-ADPr. While mono-Ser-ADPr peaks late (~20 min), as shown previously[17], mono-Asp/Glu-ADPr peaks early (~10 min) before undergoing rapid degradation (Fig. 7A and Supplementary Fig. 7). These observations were confirmed by IF (Fig. 7B). Evaluating the impact of PARG inhibition, we observed significant increases in both mono- and poly-Asp/Glu-ADPr in WT and HPF1KO cells (Fig. 7C), consistent with earlier findings (Fig. 6). Interestingly, unlike the sustained levels of poly-Asp/Glu-ADPr after 40 min, mono-Asp/Glu-ADPr declined rapidly after its initial peak (~10–20 min) (Fig. 7C, D). Therefore, DNA damage triggers similar dynamics for poly- and mono-Asp/Glu-ADPr in untreated cells, but PARGi treatment uncouples their responses. This could be partly due to other hydrolases, like TARG1 and MACROD1/2[6], eventually removing mono-Asp/Glu-ADPr.

## Discussion

The recognized biological and clinical significance of ADPr contrasts sharply with the challenges faced in studying this chemically complex and often labile modification at the molecular level. While the tools and methodologies developed by our team and others have significantly advanced research on stable forms of ADPr[22], especially Ser-ADPr, the study of more elusive types of ADPr, including on aspartate and glutamate, continues to trail considerably behind. This is despite the longstanding recognition of aspartate and glutamate as primary targets for PARP1-dependent ADPr[12,27,28,44,47]. To address this technical gap, our study has introduced preparation methods generally aimed at preserving ester-linked PTMs. We have integrated these protocols with our highly sensitive and specific SpyTag antibodies[17] as well as proteomics technology. Our results are consistent with recent reports showing that boiling the sample causes significant losses of the ADP-ribose signal[20,38]. While heating at 60 °C in a non-denaturing lysis buffer, which requires PARP inhibitor to prevent post-lysis ADPr, has been proposed as an alternative to boiling[38], a key feature of our western blotting protocol is the avoidance of heating samples. Instead, we maintain them mostly at 4 °C and occasionally at room temperature, while still ensuring efficient denaturing cell lysis that inactivates post-lysis activity of PARP1 and PARG. This principle is crucial due to the high thermolability of Asp/Glu-ADPr and allows, to a large extent, the use of standard buffers and procedures, thereby ensuring the protocol can be promptly implemented in any biological laboratory. In contrast, sample preparation for proteomics involves different considerations, as prolonged heating at 37 °C is required for effective protein digestion. Hence, for proteomics, we preserved ester-linked ADPr by establishing an acid digestion protocol.

This combined methodology has enabled us to uncover PARP1-dependent mono-Asp/Glu-ADPr in the DNA damage response (Fig. 7D). This previously elusive PTM is now clearly detectable and, in WT cells, it co-exists with Ser-ADPr, while becoming the predominant form of mono-ADPr in cells depleted of the Ser-ADPr writer HPF1. While our study focuses on mono-ADPr, in terms of amino acid target specificity, it dispels doubts about the prevalence and significance of Asp/Glu-ADPr in the DNA damage response as demonstrated by prior research[12,27,28,44,47], while confirming serine as the primary, but not the only, target in the context of HPF1 signaling.

Remarkably, this method has enabled us to demonstrate that cellular levels of mono-Asp/Glu-ADPr are regulated by PARG. This observation was unexpected since human PARG has traditionally been viewed as a hydrolase that exclusively targets poly-ADPr, and shown to be inactive against the initial protein-linked mono-ADPr unit. This has been previously demonstrated by us in the case for mono-Ser-ADPr in cells[14,17], and by others for mono-Asp/Glu-ADPr in biochemical reactions[13,15,16,18,19]. Our data suggesting that PARG can remove mono-Asp/Glu-ADPr in cells align with our subsequent finding that PARG removes mono-Asp/Glu-ADPr in biochemical reactions as well, albeit significantly less efficiently than poly-ADPr. This is supported by evidence from PARG zymograms[21], which demonstrate the complete disappearance of PARP1 automodification, as well as by a recent report showing that recombinant PARG cleaves the ester bond from mono-Asp/Glu-ADPr peptides[20]. At the same time, we cannot dismiss the possibility that the observed effects of PARG inhibition on cellular mono-Asp/Glu-ADPr levels might, at least in part, be attributed to its influence on known enzymes involved in the removal of mono-Asp/Glu-ADPr.

The ability of human PARG to cleave ester linkages is even more intriguing considering its inability to hydrolyze the O-glycosic bond between ADP-ribose and the hydroxyl group of serines[14], despite its high efficiency towards the O-glycosic bond in poly-ADP-ribose[13,18]. In contrast, *Drosophila* PARG can also hydrolyze Ser-ADPr, due to subtle structural differences in its active site[32]. Therefore, human PARG appears to have evolved to play highly specialized roles in PARP1 signaling, not only as a hydrolase of poly-ADPr and, as unveiled in this work, of mono-Asp/Glu-ADPr, but also as a 'poly-to-mono converting' enzyme responsible for forming much, though not all, of mono-Ser-ADPr[14,17].

A previous proteomics method based on hydroxylamine-based cleavage of the ester bond between ADP-ribose and aspartate/glutamate has identified thousands of poly-Asp/Glu-ADPr sites through the detection of a hydroxylamine-derived remnant mark[27,28,46]. The discovery of ADPr's prominence on other amino acids, particularly serine, has driven the development of proteomic approaches capable of detecting ADPr on targets beyond aspartate and glutamate[10,11,14,25,26,33]. The latter strategies, which are based on the direct detection of ADP-ribose on various amino acids without inducing chemical modifications, have mapped only a handful of Asp/Glu-ADPr sites. This, combined with a study showing that remnant signatures can also arise from hydroxylamine cross-reactivity on unmodified aspartates and glutamates[33], has raised questions about the actual extent of Asp/Glu-ADPr involvement in the DNA damage response, even in cells lacking the Ser-ADPr promoting factor HPF1. Here we have developed a rapid, low-pH digestion approach that leverages Arg-C Ultra, a protease set to have a considerable impact on many proteomics applications, to preserve Asp/Glu-ADPr during direct proteomic analyses. The dramatic loss of Asp/Glu-ADPr sites observed with conventional trypsin-based digestion suggests that their chemical lability could be the reason for detecting only a handful of these sites among thousands of Ser-ADPr sites in direct proteomics studies[25,26,32,33]. Importantly, while previously identified Asp/Glu-ADPr sites upon DNA damage are considered sites for poly-ADPr, our proteomics approach specifically and unambiguously detected ester-linked mono-ADPr sites. While affirming the validity of the hydroxylamine proteomics approach[27,28,46], our approach uniquely enables comprehensive analyses across a spectrum of ADPr forms, including Asp/Glu and Ser-ADPr.

More broadly, these detection methods open up the possibility to study Asp/Glu-ADPr signaling in other biological contexts. The vast majority of the 17 human PARP enzymes are capable of mono-ADP-ribosylating aspartate and glutamate[19]. Consequently, this PTM might play important roles in a wide range of biological and disease processes, including the antiviral immune response, protein homeostasis, and gene regulation. Nonetheless, our study suggests that its abundance and biological role may have been severely underestimated due

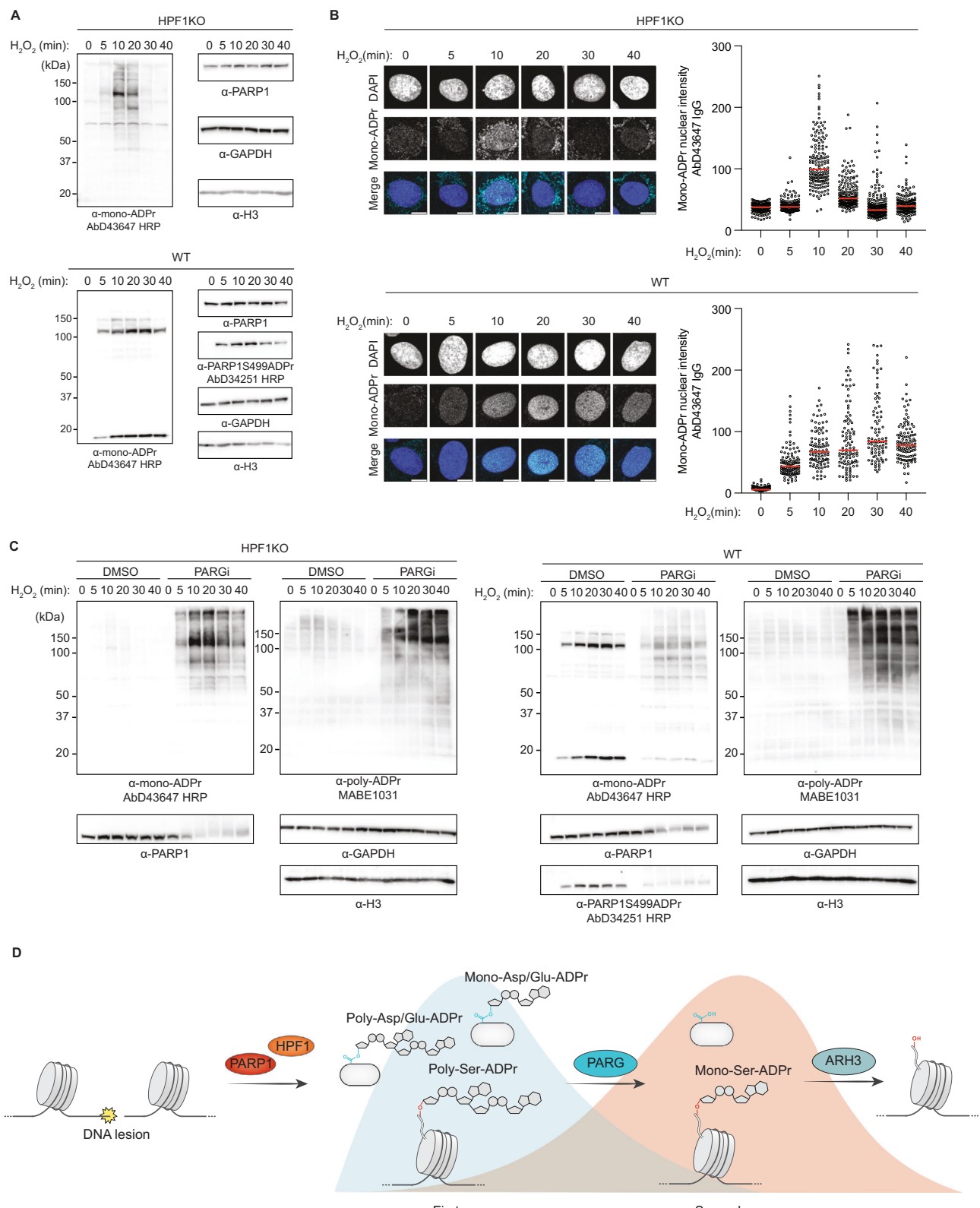

to preparation methods that systematically select against it. In this context, we anticipate that our methodology will help reveal the relative scope and specific roles of Asp/Glu-ADPr in various signaling pathways, similar to what we have achieved here for PARP1 and the DNA damage response. Importantly, as demonstrated here for models of ester-linked ubiquitination, our methodology is not restricted to ADPr, as it is in principle applicable to all known and yet-to-be discovered ester-linked PTMs. This includes serine/threonine ubiquitination, which has gained prominence with the discoveries of enzymes dedicated to this unconventional type of ubiquitination[2,3], as well as ADP-ribose ubiquitination by DELTEX E3 ligases[39].

In conclusion, this methodology promises to become highly useful in elucidating a variety ester-linked PTMs. By uncovering and providing insights into PARP1-dependent mono-Asp/Glu-ADPr in DNA

**Fig. 7 | Mono-Asp/Glu-ADPr belongs to the initial wave of PARP1 signaling.**
**A** HPF1KO (top) or WT (bottom) U2OS cells were treated with 2 mM $H_2O_2$ for the indicated time points, followed by harvesting and lysis according to our optimized procedure and immunoblotted with the indicated antibodies. See also Supplementary Fig. 7 for staining with additional antibodies. Shown is a representative result from three independent experiments. **B** Right: immunofluorescence quantification of HPF1KO (top) or WT (bottom) U2OS cells, treated with 2 mM $H_2O_2$ for the indicated time points, fixed with methanol and stained with the indicated antibody and DAPI. The intensity of ~100 cells from a representative of 3 independent experiments is shown. Red bars represent sample mean. Left: representative confocal images. Scale bar = 10 μm. **C** HPF1KO (left) or WT (right) U2OS cells were treated with 1 μM PARGi or DMSO, followed by 2 mM $H_2O_2$ treatment at

the indicated times, lysed according to our optimized procedure and immunoblotted with the indicated antibodies. Shown is a representative result from three independent experiments. **D** Simplified schematic of the main PARP1 signaling steps described in this study. Upon DNA damage, free PARP1 synthesizes mono-ADPr on aspartate and glutamate residues of several target proteins, including PARP1 itself. Mono-Asp/Glu-ADPr is, together with the known poly-Asp/Glu-ADPr, part of an initial wave of PARP1 signaling. Mono-Asp/Glu-ADPr is partly degraded by PARG and, consequently, PARG depletion or inhibition dramatically raises mono-Asp/Glu-ADPr, even in the presence of HPF1. By contrast, the PARP1:HPF1 complex switches catalytic activity towards formation of mono-Ser-ADPr, part of a second wave of PARP1 signaling and removed by ARH3. This schematic was generated in Adobe Illustrator using some graphical elements adapted from Longarini et al.[17].

damage response, we illustrate a means for accessing modifications previously rendered largely invisible and create a basis for facilitating studies of any PTMs containing ester bonds.

## Methods

### Cell culture and drug treatments
U2OS cell lines were obtained, authenticated by STR profiling and confirmed mycoplasma free by ATCC cell line authentication services. Cells were routinely tested for mycoplasma contamination. HPF1KO U2OS cells were generously provided by Ivan Ahel (University of Oxford). PARGKO U2OS cells were generously provided by Roderick J. O'Sullivan (University of Pittsburgh). Each cell line was cultured in Glutamax-DMEM supplemented with 10% bovine serum and 100 U/ml penicillin/streptomycin at 37 °C and 5% CO2. To induce PARG inhibition, the cell medium was aspirated and replaced with 37 °C complete DMEM containing 2 μM PDD00017273 for 30 min. To induce PARP inhibition, the cell medium was aspirated and replaced with 37 °C complete DMEM containing either 1 μM Olaparib, or 20 nM AZD5305, or 20 nM AZD9574 for 30 min. To induce DNA damage, the cell medium was aspirated and replaced with 37 °C complete DMEM containing 2 mM H2O2 for the indicated times.

### Immunoblotting
**Regular sample preparation.** U2OS cells (WT, HPF1KO, PARGKO) were treated as indicated in the Figure legends, then lysed in SDS buffer (4% SDS; 50 mM HEPES pH 7.9, 150 mM NaCl, 5 mM $MgCl_2$), boiled for 5 min at 95 °C, and sonicated for 10 cycles of 30 s on/off on a bioruptor (Diagenode) at 4 °C. The samples were then boiled for 5 min at 95 °C in 1 x NuPAGE LDS sample buffer (Invitrogen) with 5 mM final concentration of DTT (Sigma), resolved on NuPAGE Novex 4–12% Bis-Tris gels (Invitrogen), and transferred onto nitrocellulose membranes (Amersham) using wet transfer at 110 V for 90 min. The membranes were blocked in 5% milk in PBS buffer with 0.1% Tween-20 for 1 h at room temperature and incubated overnight with primary antibody (0.1 μg/ml for AbD43647[17], 1:1000 for commercial antibodies) at 4 °C. For antibodies requiring secondary antibody, this was followed by a 1 h incubation at room temperature with peroxidase-conjugated secondary anti-mouse (Amersham, 1:8000) or anti-rabbit (Amersham, 1:8000).

**Optimized sample preparation.** U2OS cells (WT, HPF1KO, PARGKO) were treated as indicated in the Fig. legends, lysed in SDS buffer (4% SDS; 50 mM HEPES pH 7.2, 150 mM NaCl, 5 mM $MgCl_2$), incubated for 5 min at room temperature with recombinant benzonase (smDNase, 750 U per sample). SDS is a strong denaturing agent that stops enzymatic activity in mammalian cells and obviates the need for inhibitors (see Supplementary Fig. 2a, b). Benzonase is an engineered endonuclease derived from bacteria that maintains enough activity in SDS to degrade DNA. After this stage, samples can be flash-frozen and stored at −20 °C. We did not observe significant loss in mono-Asp/Glu-ADPr signal quality after 1–2 freeze-thawing cycles. 1 x NuPAGE LDS sample buffer (Invitrogen) with 5 mM final concentration of DTT (Sigma) was added to the samples. The pH of NuPAGE LDS sample buffer is ~8.5 and

can lead to loss of Asp/Glu-ADPr if incubated for prolonged periods of time (see Supplementary Fig. 2d). Therefore, it is advisable to proceed with sample loading in <30 min, preferably within 5 min of NuPAGE addition. Afterwards, the samples were loaded on 4–12% Bis-Tris gels (Invitrogen). The operating pH of Bis-Tris gels is ~7.0 and therefore preferred over Tris-Glycine gels (operating pH of 8.3). Additionally, the running tank is kept in ice-cold water to cool the sample during the run. After SDS-PAGE, the gels are transferred onto nitrocellulose membranes (Amersham) using wet transfer at 110 V for 90 min. The transfer buffer is 1 x NuPAGE transfer buffer (Invitrogen), 20% ethanol in water. The pH of NuPAGE transfer buffer is ~7.08–7.32, thereby preserving the neutral pH established during SDS-PAGE. During this step, it's important to pre-cool the transfer buffer on ice and keep the transfer tank in ice-cold water during transfer. The membranes were then processed as described above.

The following primary antibodies were used for immunoblotting: Anti-H3 polyclonal antibody Cell Signaling Technology Cat# 9715S, Anti-PARP1 polyclonal antibody Abcam Cat# ab32138, Anti-GAPDH monoclonal antibody Millipore Cat# CB1001, anti-GFP antibody Takara Cat#632381, Anti-PARG monoclonal antibody Cell Signaling Technology Cat#66564, Anti-Ubiquitin monoclonal antibody Cell Signaling Technology Cat# 3936S, Streptavidin (anti-biotin) Cell Signaling Technology Cat#3999, Anti-mono/poly-ADP-ribose antibody (E6F6A) Cell Signaling Technology Cat# 83732, Anti-Poly-ADP-ribose binding reagent (MABE1031) Millipore Cat# MABE1031, Anti-H3S10-ADP-ribose AbD33644 Bio-Rad Cat#TZA022, Anti-PARP1S499-ADP-ribose AbD34251 Bio-Rad Cat#TZA022, Anti-Mono-ADP-ribose AbD33205 Bio-Rad Cat#TZA021, Anti-Mono-ADP-ribose AbD43647 Bio-Rad Cat#TZA020.

All commercial primary antibodies were used at 1:1000 dilution, except for anti-GFP which was used at 1:4000 dilution. AbD43647 was used at 0.1 μg/ml, AbD33644 at 0.05 μg/ml, AbD34251 at 2 μg/ml, AbD33205 at 2 μg/ml.

The following secondary antibodies were used: Anti-mouse IgG HRP-conjugated secondary Amersham Cat# NA931V, Anti-rabbit IgG HRP-conjugated secondary Merck Cat# GENA934-1ML.

### Immunofluorescence
U2OS cells (WT, HPF1KO, PARGKO) were cultured on glass coverslips, treated as indicated, and fixed with ice-cold methanol for 20 min at −20 °C. The cells were permeabilized with 0.5% Triton X-100 in PBS for 5 min, then blocked with 3% normal goat serum (Invitrogen) for 5 min. The coverslips were incubated with primary antibody (AbD43647 IgG, 1:500 dilution) for 1 h at room temperature, followed by 1 h incubation at room temperature with Alexa Fluor anti-mouse secondary antibodies (Anti-Mouse Alexa-Fluor 594-conjugated goat secondary Invitrogen Cat# A11005) at 1:500 dilution and with DAPI at 1:1000 dilution, then mounted with Prolong Diamond Antifade (ThermoScientific). Cells were imaged using a Leica SP8-DLS inverted laser-scanning confocal microscope using 63X objective. Image analysis and quantification was performed using the Fiji Software. Nuclei were identified based on DAPI, and used as a mask to measure the pixel intensity of other image channels.

## Immunofluorescence with on-slide recombinant protein treatment

For the experiment in Fig. 6E, cells grown on coverslips where fixed with methanol, permeabilized, and blocked as described above. After NGS blocking of the samples, the coverslips were incubated with or without recombinant PARG at the indicated concentrations in 500 μl of 1 x Asp/Glu-ADPr buffer (20 mM HEPES pH 7.2, 50 mM NaCl, 5 mM MgCl₂) for 2 h at room temperature. For hydroxylamine treatment, the samples were incubated with 500 μl of 1 x Asp/Glu-ADPr buffer with freshly added 1 M hydroxylamine. To control for non-enzymatic loss of Asp/Glu-ADPr during this step, all samples were incubated in the same buffer for the same time. After incubation, the coverslips were washed in PBS and incubated with primary and secondary antibodies as described above.

## Recombinant protein assays

PARP1E988Q and PARP1WT were automodified as previously described[14], using Asp/Glu-ADPr reaction buffer (20 mM HEPES pH 7.2, 50 mM NaCl, 5 mM MgCl₂). After stopping the reaction with 1 μM Olaparib, 40 nM PARP1 (E988Q or WT) was incubated with recombinant PARG at a range of time and concentration (as indicated in the respective figures), before stopping the reaction with 1 x NuPAGE LDS sample buffer and proceeding with immunoblotting as described above. For the experiment in Supplementary Fig. 6e, f automodified PARP1E988Q produced as described above was digested using the optimized method described in the "Protein digestion for mass spectrometry" section. The dried peptides were resuspended in 1 x Asp/Glu-ADPr reaction buffer and treated with or without 500 nM PARG for 2 h at room temperature. To control for non-enzymatic loss of Asp/Glu-ADPr, all samples were incubated in parallel at the same temperature for the same amount of time. Since the peptides are purified in acidic environment, it is important to ensure that the peptides are fully dried to remove formic acid, and the pH of the peptide solution should be checked to ensure it's at pH 7.0–7.2, before addition of PARG. After PARG treatments, the peptides were directly purified using stagetips as described, eluted with 30 % ACN, 0.1 % FA, dried and injected for mass-spectrometry.

ADP-ribosyl-ubiquitin on H3S10ADPr was produced as previously described[39]. Briefly, 0.5 μM UBE1, 2.5 μM UBCH5A, 10 μM Ub, 5 μM DTX2 RING-DTC, and 2 μg of H3(1-21)S10ADPr peptide were incubated at 37 °C for 30 min in 50 mM HEPES pH 7.2, 50 mM NaCl, 5 mM MgCl2, 1 mM DTT, with or without 1 mM ATP, as indicated.

## Formic acid treatment of peptides and proteins

Peptides and proteins were treated with formic acid as previously described[35]. Briefly, purified peptides stored in MS-grade water were incubated with 44 % formic acid (MS-grade) at 37 °C for 1 h. After the reaction, the peptides were directly stagetipped, dried, and analyzed by mass-spectrometry. For the ARH3 control reactions, peptides were treated with 0.1 μM ARH3 in 20 mM HEPES pH 7.0, 50 mM NaCl, 5 mM MgCl2 for 1 h at room temp, acidified with 0.1 % formic acid and processed by stagetipping. For formic acid treatment of ADPribosylated full-length Histone H3, purified recombinant Histone 3.1 was modified as previously described[14], then stagetipped, eluted with 40% ACN, 0.1% FA and dried. The dried eluate was resuspended in pure MS-grade water, or 44% formic acid in water, or in ARH3 reaction buffer, before incubation for 1 h at 37 °C (formic acid) or room temp. (ARH3). The sample was then stagetipped and dried before resuspension in 1X NuPage LDS and immunoblotted.

## Co-immunoprecipitation

U2OS cells were transfected with GFP-PARP1 (1:3 DNA:PEI ratio in Optimem, Gibco) for 24 h then treated with 2 mM H2O2 for 10 min. After two washes in ice-cold PBS, the cells were harvested by scraping and recovered by centrifugation at 500 x g for 5 min at 4 °C. For each

pulldown, a number of cells equivalent to a confluent 10-cm dish was used. The cell pellet was resuspended in 200 μl of lysis buffer (20 mM HEPES pH 7.0, 300 mM NaCl, 2.5 mM MgC2, 0.5% NP-40, 20 % glycerol, 750 U/ml benzonase, 1 μM Olaparib, 1 μM ADP-HPD, 1X EDTA-free protease inhibitor cocktail) and incubated with shaking (1400 rpm) for 30 min at 4 °C. After quenching the benzonase with 200 μl of 20 mM HEPES pH 7.0, 30 mM EDTA, 0.5% NP-40, 1 μM Olaparib, 1 μM ADP-HPD, 1X EDTA-free protease inhibitor cocktail, the lysate was clarified by centrifugation at 16,000 xg for 5 min at °C, followed by collection of the supernatant. The supernatant was incubated with washed GFP-trap M-270 magnetic beads (Chromotek) for 1 h at 4 °C, 1400 rpm. 25 μl of beads were used for each pulldown. After incubation, the beads were washed first with 3 washes in 20 mM HEPES pH 7.0, 300 mM NaCl, 2 % NP-40, 0.5 mM EDTA then 3 washes in 8 M urea, 20 mM HEPES pH 7.0, 1 mM DTT. The washes in urea buffer remove non-specific and specific interactors of GFP-PARP1, thereby generating a cleaner elution to improve the detection of low-stoichiometry ADPr sites. Afterwards, the proteins were eluted by digestion and processed for mass-spectrometry as indicated in the "Protein digestion for mass spectrometry" section.

## Protein digestion for mass spectrometry

Two different sample preparation methods for mass-spectrometry were employed in this study (Fig. 3A). In the first, we used a standard in-solution trypsin digestion in guanidinium buffer at slightly basic pH for overnight at 37 °C, conditions which are representative of a routine digestion protocol but lead to artefactual hydrolysis of ester-linked PTMs (see Supplementary Fig. 3a). In the second we optimized digestion conditions to preserve protein-linked Asp/Glu-ADPr which can be achieved by short incubations at low pH (pH ~ 5.0). Trypsin digestion is not suited for acidic pH. To circumvent this limitation, we used the recently introduced Arg-C Ultra (Promega), which shows increased specificity and efficiency compared to regular Arg-C and preserves high activity at low pH (https://www.promega.com/-/media/files/promega-worldwide/europe/promega-germany/massspec/ps490-asms203-hosfield.pdf?sc_lang=en?utm_source=newsletter&utm_medium=email&utm_campaign=de-2023-massspec#msdynttrid=sfcOB56vCS8k2qLwO3PY0TOJmTikMmh3OKkvvpo6KSc). To complement Arg-C Ultra we included Lys-C (Mass-spectrometry grade, Fujifilm Wako), which preserves activity at slightly acidic pH.

For experiments with recombinant PARP1E988Q, 4 μg of protein modified as described in the Recombinant protein assays section were denatured with the addition of equal volumes of 8 M urea, 20 mM sodium acetate (pH 5.0), 10 mM TCEP and incubated for 15 min at 37 °C. Then, the samples were diluted 10-fold in Arg-C Ultra digestion buffer (20 mM sodium acetate pH 5.0, 10 mM TCEP) and incubated for 3 h at 37 °C with 1:5:20 μg ratio of Arg-C Ultra:Lys-C:Protein. The same procedure was followed for digestion of GFP-PARP1 immunoprecipitated from cells, in which case 0.25 μg of Arg-C Ultra and 1 μg of Lys-C was used for each sample. The digestion was stopped with the addition of 2% formic acid (FA) and processed with C18 stagetips according to a standard protocol with 100% MeOH conditioning buffer, 30% ACN 0.1% FA equilibration and elution buffer, 0.1% FA washing buffer. The eluate was dried to completion in a speedvac (Eppendorf). The dried peptides were stored at −20 °C until resuspension in 0.1% FA prior to injection in the mass-spectrometer.

## Enrichment of mono-ADP-ribosylated substrates

**Sample lysis and digestion.** U2OS HPF1KO cells were treated with 2 mM H2O2 for 10 min, harvested in PBS and pelleted by centrifugation at 500 g for 5 min. We processed the cell pellets in two ways. In one set of experiments, we performed subcellular fractionation to enrich the nuclear soluble fraction. In another, we processed the samples as a whole cell lysate.

**Nuclear soluble fraction enrichment.** The cell pellet was resuspended in hypotonic buffer (10 mM HEPES pH 7.0, 10 mM KCl, 0.1 mM EDTA, 0.1 mM EGTA, supplemented with fresh 1 mM DTT, 0.5 mM PMSF, 10 μM olaparib, 1 μM ADP-HPD, and EDTA-free protease inhibitors cocktail) and left on ice for 15 min. After incubation, 1:50 detergent (Active Motif #40010) was added and the tubes vortexed for 10 s before pelleting by centrifugation at 14000 g for 30 sec. The pellet was resuspended in nuclear extraction buffer (20 mM HEPES pH 7.0, 420 mM NaCl, 2.5 mM MgCl$_2$, 0.01% NP-40, 20% glycerol, supplemented with fresh 10 μM olaparib, 1 μM ADP-HPD, and EDTA-free protease inhibitors cocktail) and incubated for 30 min at 4 °C with shaking. The samples were then centrifuged at 14,000 for 5 min and the supernatant, containing nuclear soluble proteins, was processed further. The pellet was flash frozen and stored at −80 °C as nuclear pellet fraction. Proteins in the nuclear soluble fraction were precipitated by addition of 4-fold volume of ice-cold acetone and incubated overnight at −20 °C. After incubation, the samples were centrifuged for 5000 g for 20 min, the supernatant was discarded and the pellet washed with 2 mL of ice-cold acetone. This procedure was repeated twice. The final pellet was left to air dry until all traces of acetone evaporated. The pellet was resuspeneded in 8 M urea, 20 mM HEPES pH 7.0, 5 mM DTT and incubated at 37 °C for 10 min. CAA was addded to a final concentration of 15 mM and incubated for 1 h in the dark at room temperature. The CAA was quenched by addition of 15 mM DTT and incubation for 10 min at room temperature. At this stage, the protein concentration was estimated by nanodrop, with 10-fold dilution to dilute DTT, and a volume corresponding to 10 total mg of proteins was processed further. The samples were diluted 10-fold in 20 mM sodium acetate (pH 5.0), 5 mM DTT. Arg-C Ultra was added to the samples at 1:2000 protease:protein ratio (w/w) and incubated for 3 h at 37 °C. After digestion, the sample was acidified with 2 % final concentration of formic acid, centrifuged at 5000 g for 5 min to remove insoluble material and the superatant was desalted with C18 Sep Pak Vac 3cc (500 mg) cartridges followed by peptide elution with 30 % ACN, 0.1 % FA. The eluted peptides were flash frozen in liquid nitrogen and dried on a lyophilizer until completely dry (~24 h).

**Whole cell lysis.** The cell pellet was resuspended in urea lysis buffer (8 M urea, 20 mM HEPES pH 7.0, 5 mM DTT, 5 mM MgCl$_2$ with 10 μM olaparib, 1 μM ADP-HPD, EDTA-free protease inhibitor cocktail), 750 U benzonase was added to each sample and incubated at room temperature for 10 min with sample agitation (1400 rpm) and for a further 5 min at 37 °C. An aliquote corresponding to 10 mg of proteins was taken. CAA was added to a final concentration of 15 mM followed by incubation for 1 h in the dark. DTT was added to a final concentration of 15 mM for 15 min at room temp. The sample was diluted 10-fold in digestion buffer (20 mM sodium acetate pH 5.0, 5 mM DTT), Arg-C Ultra was added at a 1:2000 ratio (protease:protein w/w) and incubated for 3 h at 37 °C followed by acidification and peptide purification and liophylization as described above.

**Enrichment of mono-ADPr peptides.** The peptides were resuspended in IP buffer (50 mM MOPS pH 7.2, 20 mM sodium phosphate, 50 mM NaCl) then pelleted by centrifugation at 16000 g for 5 min to remove insoluble material. The supernatant was incubated with 50 μg of anti-mono-ADPr AbD43647 IgG antibody for 2 h at 4 °C on an orbital shaker. Meanwhile, 20 μg of Protein A Agarose beads (Cell Signaling Technology) were washed twice in IP buffer using low-bind tubes (Eppendorf) by pelleting by centrifugation at 2000 g for 30 sec. After antibody incubation, the sample was added to the washed beads and incubated further for 1 h at 4 °C on an orbital shaker. After incubation, the beads were washed three times with ice-cold IP buffer and two times with ice-cold HPLC water. After the third and fifth wash step, the supernatant was transferred to clean low-bind tubes. Peptide elution was carried out by incubation for 5 min with 600 μl of 0.15 % TFA at

room temperature, pelleting by centrifugation at 2000 g for 30 sec. The elution step was repeated for a total of three times. The eluate was split and loaded in several stagetips with enough capacity to account for co-elution of the antibody (e.g. three 30 μg-capacity stagetips for 50 μg of eluted antibody), the sample was desalted according to standard stagetip procedures and eluted with 30 % ACN, 0.1 % FA. The eluted peptides were dried in a speedvac and resuspended in 0.1 % FA prior to mass-spectrometry analysis.

## LC-MS/MS data acquisition

For the identification and localization of Asp/Glu-ADPr peptides, three different data acquisition methodologies were used in this study. Briefly, two methodologies rely on ultra-fast MS/MS with HCD to detect the intense adenine diagnostic peak (mass = 136.061) and trigger the subsequent acquisition of high-quality HCD or ETD spectra, respectively, while the third method is exclusively high quality ETD on all precursors. In practive, Data-Dependent Acquisition (DDA) data were collected by a TopN ($N = 20$) method on the Orbitrap Fusion LUMOS system (Thermo Scientific) equipped with a FAIMS-Pro interface and coupled to an Easy-LC 1200 with a 75 micron x 40 cm fused silica capillary (CoAnn Technologies) packed with Poroshell 120 EC-C18 2.7 micron medium (Agilent). MS1 spectra were acquired at 60 k resolution, 300% normalized AGC, 50 ms maximum injection time, alternating between three FAIMS CVs (−40, −50, −60).

**Triggered ETD.** Precursors with charge states 3–9 (highest first) and adenine diagnostic ion were further re-isolated and fragmented by ETD. MS2 spectra were collected at 60 k resolution with AGC target set to "standard" and maximum injection time set to "auto".

**Triggered HCD.** Precursors with adenine diagnostic ion were further re-isolated and fragmented by HCD. MS2 spectra were collected at 60 k resolution with normalized AGC target set to 400% and maximum injection time set to 250 ms.

**Pure ETD.** Precursors with charge states 3–9 (highest first) were fragmented by ETD. MS2 spectra were collected at 60k resolution with AGC set to 400% and maximum injection time set to 120 ms.

## Quantification and statistical analysis

All data analysis was performed using GraphPad Prism 9, ImageJ, Microsoft Excel and R (RStudio version 1.4.1717). For the quantification of microscopy data, a custom ImageJ script was used to segment nuclei based on DAPI signal and apply as a mask to measure the intensity of other channels. For mass-spectrometry samples,.raw files were processed using MaxQuant[61], version 1.5.2.8, and further analyzed with custom R scripts. The data were searched against a human PARP1 FASTA file with the following parameters. The digestion mode was set to trypsin and the maximum number of missed cleavages was set to 6. Re-quantify and match-between-runs was enabled. Variable modifications included oxidation (M), acetylation (protein N-term) and ADP-ribosylation (DEKRSTC). The ADPr modification allowed for AMP neutral loss (AMP diagnostic peak m/z 348.0704). For stringent identification of ADPr sites we filtered the *ADPr Sites* output for delta score > 40 and localization probability ≥ 0.9.

## Use of Large Language Models (LLMs) in the writing process

During the preparation of this work the authors used ChatGPT 4 in order to improve readability and language. After using this tool/service, the authors reviewed and edited the content as needed and take full responsibility for the content of the publication.

## Reporting summary

Further information on research design is available in the Nature Portfolio Reporting Summary linked to this article.

## Data availability

Mass spectrometry data have been deposited in the ProteomeXchange Consortium (http://proteomecentral.proteomexchange.org) with the dataset identifier ProteomeXchange: PXD048274. Source data are provided with this paper.

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

## Acknowledgements
We are grateful to Daniel Squair and Satpal Virdee from the University of Dundee for the kind gifts of the E2–Ub model substrate. We thank Ivan Ahel and Kang Zhu (Oxford University) for providing recombinant DTX2 and PARP1E988Q. PARG U2OS KO cells were a kind gift from O'Sullivan laboratory, University of Pittsburgh. We thank the MPI AGE Proteomics Core Facility, in particular Thomas Colby, Ilian Atanassov and Xinping Li, for help with mass spectrometric analyses. We also thank the MPI AGE FACS and Imaging Facility for help with immunofluorescence experiments. This work was funded by the Max Planck Society, the Deutsche Forschungsgemeinschaft (DFG, German Research Foundation) under Germany´s Excellence Strategy (CECAD, EXC 2030-390661388) and by the European Research Council (ERC-CoG-864117) to I.M. E.J.L. received support by the Cologne Graduate School of Ageing Research. I.M. was supported by the EMBO Young Investigator Programme.

## Author contributions
I.M. and E.J.L. conceived the project, designed research and wrote the manuscript. E.J.L performed all the experiments.

## Funding

## Competing interests
The authors E.J.L. and I.M. declare the following competing financial interest: Max-Planck-Innovation, which is responsible for technology transfer from Max Planck Institutes, has licensed the antibodies AbD33205, AbD33644, AbD34251 and AbD43647 to Bio-Rad Laboratories, which markets them for research purposes.
