## [Peer Review File · Nature Communications]

Reviewers' Comments:

Reviewer #1:

Remarks to the Author:

The manuscript by Longarini and Matic describes the development of an improved protocol that finally retains ADP-ribose on Asp/Glu residues for both immunoblots and mass-spectrometric analysis. They also show results that the modified protocol should be also useful to identify other post-translational modifications attached to the amino acid residue through an ester bond such as ubiquitination of ADP-ribose. Using this improved protocol, they show that PARG can also remove ADP-ribose from Asp/Glu residues. Undoubtedly, the modified protocol should greatly facilitate the research of ester linked modifications, therefore, it is a very important finding that should be published without much delay.

Minor comments:

On P4L55 authors write that: "...according to the current consensus, is unable to remove the initial protein-ribose bond, leaving a mono-ADPr remnant on its targets." Yet later, on P14L370 they write that "a recent report suggesting that recombinant PARG can remove mono-Asp/Glu-ADPr from in vitro modified peptides" citing Ref27. – I suggest discussing ref 27 already in the introduction.

I did not find any information if during cell lysis PARP and PARG inhibitors are present or not. They are not mentioned being part of the lysis buffer. If they were not included during lysis, authors should show data that they are not required because if PARP1 would not be fully denatured in the lysis buffer and retains some activity then the DNA fragments formed by adding benzonase could cause strong PARP1 activation.

It would be interesting to compare that protein targets of PARP1 in the presence and the absence of HPF1. This might reveal how PARP1 selects its substrates, and how much it is shifted when HPF1 is not present.

Similarly, using the power of mass spectrometry, authors should try to address if certain characteristics might be revealed about ADPr-ylated Asp/Glu residues from which PARG can efficiently remove ADPr and about those on which PARG appears inefficient.

Reviewer #2:

Remarks to the Author:

The manuscript by Longarini and Matic describes an optimized protocol for detecting labile Glu/Asp-mono-ADP-ribosylated proteins. They also show that inhibition of PARG in cells leads to an increase in Glu/Asp-mono-ADP-ribosylated proteins. This result, coupled with in vitro experiments showing that PARG (albeit at high concentrations), led the authors to conclude that PARG is a Glu/Asp-mono-ADP-ribose hydrolase.

The manuscript generally exhibits good writing quality, and the experiments are adequately controlled. The optimized protocol for detecting Glu/Asp-mono-ADP-ribosylated proteins, especially by mass spectrometry, is an important contribution to the field. While the PARG inhibitor and KO results certainly point to PARG being a potential Glu/Asp-mono-ADP-ribose hydrolase, the high concentration of PARG required to remove Glu/Asp-mono-ADP-ribose in vitro raises some doubts about this assertion. The authors should address the comments below before publication. I've included some specific comments below.

1. The authors should reference a recent paper (Weixler et al., Life Science Alliance, 2022) that described the optimization of detecting ADP-ribosylation in cells. This paper showed that boiling the samples leads to a significant loss of the ADP-ribose signal, consistent with the current study.
2. A common strategy to confirm PTM of a target is to mutate the identified sites. Did the authors try to mutate identified glutamates in PARP1?
3. The study would be strengthened by showing that Glu-mono-ADP-ribose sites can be identified on proteins beyond the highly abundant PARP1.
4. Removal (albeit incomplete) of Glu/Asp-mono-ADP-ribose on PARP1 requires micromolar

concentrations of PARG and several hours. The concentration of PARG in cells is certainly much lower than this. How do the authors reconcile these discrepancies? Is there an alternative model for the effects of PARG inhibition in cells? For example, could PARG inhibition (or KO) impact the activity of known enzymes that remove Glu/Asp-mono-ADP-ribose?

Reviewer #3:

Remarks to the Author:

This manuscript presented by Longarini and Matic is essentially based on two observations. Firstly, the study rehabilitates PARP-1 in its physiological role of ADP-ribosylation of D/E residues. Secondly, it clarifies the eraser activity of PARG towards the proximal ADP-ribose or the mono-ADP-ribosylation of D/E residues. The study is extremely relevant because these two concepts have been the subject of controversy in the field. The results presented could lead us to reconsider or reinterpret several previous studies. With this in mind, the research theme is more than appropriate and the results obtained will be of interest to a wide range of researchers involved in the study of ADP-ribosylation. On the other hand, the weakness of the manuscript lies in the little attention that the authors make of this controversy and of the previous results presented by other teams which, precisely, support and validate the present observations. It is inconceivable that pioneering studies associated with D/E ADP-ribosylation are not mentioned, discussed, or put into context by the authors. Once again, I mention that this is an original and well-executed study but, in most part, results were expected, particularly in light of previous studies. We would rather have expected a comparison with the numerous ADP-ribosylation sites identified on the D/E residues of PARP-1 both in a cellular context and with recombinant PARP-1. In its current form, the paper is rather technical with a demonstration limited to PARP-1 without proteome-wide application. It remains relevant but perhaps too preliminary for a publication in Nature Communications. Considering that Matic's group and colleagues originally established serine as the primary form of ADP-ribosylation in DNA damage signaling (some even going so far as to speak of artifacts in relation to ADP-ribosylation on D/E residues), we would have expected a more in-depth context. I am thinking in particular of all the work of Adamietz and Hilz on the lability of ester bonds with ADP-ribose. At that time, the vast majority of studies focusing on the nature of the ADP-ribose linkages provided evidence that ADP-ribosylated histones and PARP-1 itself were primarily modified via hydroxylamine-sensitive ester type of bonds. These explanations are necessary for the reader to understand why the present study is important. Again, Matic and colleagues contributed to establish serine as the primary form of ADP-ribosylation in DNA damage signalling (Palazzo et al. Elife 2018). The manuscript should be restructured and balanced to better appreciate the contribution of each type of modification to the DNA damage response.

Major points:

1- The idea of preserving ester bonds by developing a sample preparation pipeline in acidic conditions is original and entirely appropriate. The authors explain that ester bonds are particularly likely to be hydrolyzed in mild basic conditions, hence the idea of carrying out lysis and digestion in acidic conditions. While the concept is certainly applicable to proteomic pipelines, it is unclear how acidic conditions are optimally preserved for detection of ADP-ribosylation by Western blotting. For example, in the "optimal sample preparation" method, cell extracts are loaded with 1X NuPAGE LDS sample buffer. According to the manufacturer, the pH of this solution is 8.5, which is basic enough to hydrolyze very labile ester ADP-ribose conjugates. Technically, all buffers, including gel running buffer and membrane transfer buffer should also be acidic to preserve ester modifications. These conditions are not clearly described. This is crucial as it impacts all blots presented in the manuscript.

2- As shown by the authors, acidic conditions (e.g. 44% formic acid) do not hydrolyze SER-ADP-ribosylation. Therefore, all major ADP-ribosylation linkages should be preserved in acidic conditions (i.e. carboxyl-ester (D/E) and acetal linkages on S/T). However, the authors did not report site-specific SER-ADP-ribosylation in the present study. They explained that HCD fragmentation of Asp/Glu-ADPr peptides leads to the preferential breakage of the diphosphate group of ADP-ribose, leading to the loss of AMP and presence of a phosphoribose-H₂O remnant on the modified amino-acid. This behavior is in contrast to Ser-ADPr peptides, where HCD fragmentation leads largely to complete loss of the modifier, leaving no localization-specific ions. The group of Matic have

extensive expertise on the identification of SER-ADP-ribosylation (e.g. by HCD/ETD MS fragmentation) so it is surprising to note that the new method optimized in acidic conditions has not been revalidated via their proteomic platform for SER-ADP-ribosylation. They could have shown that the new technique could be applicable to carry out exhaustive profiling of PARP-1 self-modification. The D/E-specific profiling shown in Fig 3C cannot identify the exact positions of ADP-ribosylation and is therefore unusable in its current form.

Using the raw MS data deposited on ProteomeXchange (PXD048274) we were able to validate the D/E-ADP-ribosylation HCD fragmentation pattern observed by the authors. However, the major SER-ADP-ribosylation sites are located at S499/S504/S508 and not covered by the ARG-C/Lys-C digestion conditions used in this study. The presence of semi-tryptic peptides or partial digestions are necessary to map this region and verify that the HCD fragmentation pattern is truly different between D/E-ADP-ribosylation and SER-ADP-ribosylation. Indeed, the reanalysis of the spectral data by our MS analysis software indicates site-specific ADP-ribosylation on serine residues while clearly observing the presence of the diagnostic ions adenine and AMP. On the other hand, it is difficult to find a tryptic peptide lacking D/E residues to confirm the HCD fragmentation pattern on a specific serine. Validation of the single "acid" approach used for comprehensive identification of all sites should be undertaken.

3- The authors report ADP-ribosylation sites at E168/169, E190, E471, E488, E491, E513/514 and E642 on PARP-1 in cells. Sites like E190, E488, E491 or E642 were already reported as major ADP-ribosylation sites by the group of Y. Yu (Nature Methods 2013, Cell rep 2017), L. Kraus (Science 2016), G. Poirier (NAR 2012, J Proteome Res 2018), Leung AK (J Proteome Res 2014) and back to Tao et al. (JACS 2009). The authors should explain that the sites they identified had already been validated in cells and with the use of recombinant PARP-1, which validates previous approaches and invalidates the hypothesis of artifactual changes induced by derivatization of the modification as previously reported, notably by the group of ML Nielsen (Hendriks IA et al. Nature Communications 2021).

4- The schematic model is not really representative of the biochemistry of PARP-1 during DNA damage linked to ADP-ribosylation on residues D/E and S. We do not see the two waves of ADP-ribosylation nor the period where the two modifications could be co-occurring. The authors mention that the formation of mono-Asp/Glu-ADPr is not limited to HPF1 KO cells but also occurs in WT cells. This should be part of the schematic model.

5- The characterization of PARG as a D/E mono-ADP-ribosyl hydrolase is one of the most interesting result of this study, especially the observation that PARG activity towards mono-D/E-ADP-ribosylation is prominent in cells. The authors mentioned that the rapid hydrolysis of PAR by PARG on automodified PARP-1 is followed by a slower and incomplete removal of PARG. They should mention that PARG zymograms performed with ³²P-labeled automodified PARP-1 show a complete disappearance of the radiolabeled moiety (Winstall E. et al. Exp Cell Res. 1999), suggesting a complete removal of the ADP-ribose, which is consistent with the activity observed in cells. Thus, it is possible to observe total erasing depending on the experimental conditions. Zymograms should be performed to support their results.

Minor points:

1- The authors mentioned: Although core histones are abundantly ADP-ribosylated on serine residues during DNA damage, and on glutamate in other biological contexts, we did not observe Asp/Glu-ADPr on histones under our experimental conditions. They should cite Karch KR et al. (Mol Biosyst 2017). Their data suggest that specific D/E residues are present on the nucleosomal surface.

2- In the discussion, the authors imply that the derivatization methods used in MS to identify ADP-ribosylation sites are less valid than the method directly targeting the modification itself. On the other hand, if we consider that the hydroxylamine approach developed by Yu's group to identify ADP-ribosylated D/E residues in cells has essentially been validated by the present approach (major D/E ADP-ribosylation on PARP-1 were identified in addition to thousands of other sites in hundreds of proteins), the two approaches may appear complementary. Moreover, MS signatures

generated after derivatization is a common approach in PTM analysis by mass spectrometry. There are obvious advantages to generating diagnostic ions for the analysis of ADP-ribosylated sites but the authors should nevertheless acknowledge the convergence of the two methods.

Thursday, March 21, 2024

Point-by-point answer to reviewers:

NCOMMS-24-02189-T (Preservation of Ester-linked Modifications Reveals Glutamate/Aspartate mono-ADP-ribosylation by PARP1 and its Reversal by PARG)

Summary:

We would like to thank the editor and the reviewers for their time in reviewing our work and for their helpful and constructive suggestions. We are encouraged by the generally positive and, to some extent, enthusiastic assessment of our manuscript.

All reviewers agree that our study is important, and have raised valid points, focused on strengthening specific aspects of the manuscript. We have addressed these concerns by adding substantial new data and expanding our discussion and references to include early pioneering work as well as other studies on Asp/Glu-ADPr.

Briefly, we have obtained the following additional data:

- identification of targets beyond PARP1: the first application of the AbD43647 antibody in proteomics revealed over 150 mono-Asp/Glu-ADPr sites, with 114 of these being newly identified, across a range of targets, including histones, HMG proteins, ribonucleoproteins and topoisomerases. Among these, 18 proteins represent new mono-Asp/Glu-ADPr targets at the protein level.
- detection of both Ser- and Asp/Glu-ADPr sites in the same proteomic analysis
- further investigation of PARG's role as a hydrolase of mono-Asp/Glu-ADPr
- mutation of PARP1 Asp/Glu-ADPr sites
- evidence that PARP1 and PARG are inactive in our denaturing 4% SDS lysis buffer even without boiling or the use of inhibitors
- further investigation of conditions that preserve or remove Asp/Glu-ADPr

We believe that by following reviewers' suggestions and incorporating considerable new results and clarifications, we have significantly improved the manuscript and addressed their concerns. We hope that the revised version will now be accepted for publication in *Nature Communications*. This would make our improved methods widely available, significantly advancing research of ester-linked modifications.

Please find below a detailed point-by-point answer to the referees' comments.

Table: Summary of Changes To figures

Previous version	Revised Version	
Fig. 3	Fig. 3	
Fig. 3C	[Fig. S3D]	
Fig. S3D	Fig. 3C	Revised
Fig. 3E	Fig. 3D	
Fig. 3D	Fig. 3E	Revised
-	Fig. 4	
	Fig. 4A	New
	Fig. 4B	New
	Fig. 4C	New
	Fig. 4D	New
	Fig. 4E	New
	Fig. 4F	New
	Fig. 4G	New
	Fig. 4H	New
Fig. 4	Fig. 5	
Fig. 5	Fig. 6	
Fig. 5E	[Fig. S6A]	
Fig. 5F	[Fig. S6D]	
	Fig. 6E	New
Fig. 6	Fig. 7	
Fig. 6D	Fig. 7D	Revised

Overview of manuscript changes:

Previous version	Revised Version	
Fig. S2	Fig. S2	
	Fig. S2A	New
	Fig. S2B	New
Fig. S2A	Fig. S2C	
	Fig. S2D	New
Fig. S2B	Fig. S2E	
Fig. S2C	Fig. S2F	
Fig. S2D	Fig. S2G	
Fig. S3	Fig. S3	
Fig. 3C	Fig. S3D	Revised
	Fig. S3E	New
	Fig. S3F	New
	Fig. S3G	New
-	Fig. S4	
	Fig. S4A	New
	Fig. S4B	New
	Fig. S4C	New
-	Fig. S5	
	Fig. S5A	New
	Fig. S5B	New
	Fig. S5C	New
Fig. S4	Fig. S6	
Fig. 5E	Fig. S6A	
Fig. S4A	Fig. S6B	
Fig. S4B	Fig. S6C	
Fig. 5F	Fig. S6D	
	Fig. S6E	New
	Fig. S6F	New
Fig. S5	Fig. S7	

We have modified the format of certain plots to comply with the editorial guidelines of *Nature Communications*. All changes in the manuscript, including text reductions to meet length requirements, are highlighted in yellow.

Reviewer #1 (Remarks to the Author):

The manuscript by Longarini and Matic describes the development of an improved protocol that finally retains ADP-ribose on Asp/Glu residues for both immunoblots and mass-spectrometric analysis. They also show results that the modified protocol should be also useful to identify other post-translational modifications attached to the amino acid residue through an ester bond such as ubiquitination of ADP-ribose. Using this improved protocol, they show that PARG can also remove ADP-ribose from Asp/Glu residues. Undoubtedly, the modified protocol should greatly facilitate the research of ester linked modifications, therefore, it is a very important finding that should be published without much delay.

Many thanks to the Reviewer for their highly positive assessment of our work, clear and constructive advice for improvements, and the view that our improved methodology represents 'a very important finding that should be published without much delay'. We certainly agree that our manuscript is timely, offering immediate practical utility and benefiting the field by resolving the ongoing controversy regarding the significance of Asp/Glu ADP-ribosylation. We also thank the reviewer for highlighting the broad applicability of our protocols within the wider context of ester-linked PTMs.

Minor comments:

On P4L55 authors write that: "...according to the current consensus, is unable to remove the initial protein-ribose bond, leaving a mono-ADPr remnant on its targets." Yet later, on P14L370 they write that "a recent report suggesting that recombinant PARG can remove mono-Asp/Glu-ADPr from in vitro modified peptides" citing Ref27. – I suggest discussing ref 27 already in the introduction.

We agree that it is better to discuss ref 27 in the introduction.

We have addressed this point by incorporating both the original ref 27 and reviewer #3's suggestion to cite Winstall E. et al., Exp Cell Res., 1999, as early evidence of PARG's complete removal of PARP1 ADPr. Specifically in the introduction, following the first sentence quoted by the reviewer, we have cited both papers and added 'However, this view has been recently challenged by evidence that recombinant PARG

can cleave mono-ADPr from peptides. This is supported by an earlier finding that PARG can completely degrade poly-ADPr, including the initial ADP-ribose moiety'. Papers suggesting PARG's inability to hydrolyze the initial protein-ribose bond, previously cited in later sections of the manuscript, are now also cited in the introduction.

I did not find any information if during cell lysis PARP and PARG inhibitors are present or not. They are not mentioned being part of the lysis buffer. If they were not included during lysis, authors should show data that they are not required because if PARP1 would not be fully denatured in the lysis buffer and retains some activity then the DNA fragments formed by adding benzonase could cause strong PARP1 activation.

We thank the Reviewer for pointing this out, as it is an important consideration given that we have developed a new cell lysis protocol that omits the usual boiling step. Indeed, lysis under non-denaturing conditions, such as with RIPA-based buffers, strongly activates PARP1, as shown by Weixler et al., Life Science Alliance. However, the presence of 4% SDS alone in our modified lysis buffer, aimed at preserving ester bonds, constitutes a strong denaturing condition. In our lab, we routinely use 4% SDS to quench *in vitro* reactions, such as PARP1 automodification, and therefore we did not initially consider to validate this point. However, we agree with the reviewer that this is an important aspect in the context of our manuscript that should be substantiated. As demonstrated in the new Supplementary Fig. 2a, 4% SDS without boiling is sufficient to completely inactivate both recombinant PARP1 and PARG and, accordingly, we did not observe a change in the ADPr signal from cells when PARP1 and PARG inhibitors are included in the lysis buffer (new Supplementary Fig. 2b). Therefore, PARP1/PARG inhibitors are not required. We have added the following sentences 'Non-denaturing lysis conditions can result in the post-lysis activity of enzymes, including PARP1, requiring the use of inhibitors. Crucially, 4% SDS at room temperature is sufficient to completely inactivate PARP1 and PARG (Supplementary Fig. 2a). Therefore, our cell lysis protocol does not require the use of inhibitors (Supplementary Fig. 2b).'

It would be interesting to compare that protein targets of PARP1 in the presence and the absence of HPF1. This might reveal how PARP1 selects its substrates, and how much it is shifted when HPF1 is not present.

We agree that it would be interesting to compare the protein targets of PARP1 with and without HPF1. To this end, we have performed a proteome-wide identification of mono-Asp/Glu-ADPr residues. We refer to our response for reviewer #2 for a better overview of this new experiment.

In this context, to understand how the protein targets of PARP1 differ in the presence or absence of HPF1, we have mapped the high-confidence Asp/Glu-ADPr sites identified in our new experiment to their corresponding proteins and compared across published studies which comprehensively identified Ser-ADP-ribosylated proteins. We found that PARP1 generally targets a conserved set of proteins and its target substrates are therefore largely independent of HPF1. This is with the exception of core histones, which are predominantly Ser-ADP-ribosylated, although we could still identify low-abundance histone Asp/Glu-ADPr, in particular H2BE2ADPr, confirming previous reports (Ogata et al., JBC 1980; Burzio et al., JBC 1979; Grundy et al. Nat Comms 2016). Therefore, in terms of substrate selection, the main role of HPF1 is to shift towards ADP-ribosylation of core histones, while most other targets, at the protein level, are largely unaffected. This is consistent with the observation that HPF1-dependent chromatin relaxation is largely driven by histone ADPr specifically (Smith et al., NSMB, 2023). We have added the following sentence in the manuscript: “The high confidence mono-Asp/Glu-ADPr sites mapped to 69 protein targets, the majority of which were previously identified in the context of Ser-ADPr, indicating the homogeneous nature of PARP1 substrate targeting.”

Similarly, using the power of mass spectrometry, authors should try to address if certain characteristics might be revealed about ADPr-ylated Asp/Glu residues from which PARG can efficiently remove ADPr and about those on which PARG appears inefficient.

To reply to this comment, we applied recombinant PARG to ADPr peptides that were generated through acidic digestion of automodified PARP1. Our analysis showed that PARG does not exhibit a clear preference for removing mono-ADPr from specific Asp/Glu residues, though we noted marginally more efficient removal in some peptides (new Supplementary Figs. 6e, f).

Reviewer #2 (Remarks to the Author):

The manuscript by Longarini and Matic describes an optimized protocol for detecting labile Glu/Asp-mono-ADP-ribosylated proteins. They also show that inhibition of PARG in cells leads to an increase in Glu/Asp-mono-ADP-ribosylated proteins. This result, coupled with in vitro experiments showing that PARG (albeit at high concentrations), led the authors to conclude that PARG is a Glu/Asp-mono-ADP-ribose hydrolase.

The manuscript generally exhibits good writing quality, and the experiments are adequately controlled. The optimized protocol for detecting Glu/Asp-mono-ADP-ribosylated proteins, especially by mass spectrometry, is an important contribution to the field. While the PARG inhibitor and KO results certainly point to PARG being a potential Glu/Asp-mono-ADP-ribose hydrolase, the high concentration of PARG required to remove Glu/Asp-mono-ADP-ribose in vitro raises some doubts about this assertion. The authors should address the comments below before publication. I've included some specific comments below.

We are pleased that the Reviewer considers our improved protocol an important contribution to the field. We thank the Reviewer for noting that our experiments are adequately controlled and that the writing quality of the manuscript is good.

We acknowledge that our original experiment, involving the removal of in vitro automodified PARP1 with a large amount of recombinant PARG, introduces some doubts about PARG's role as a Glu/Asp-mono-ADP-ribose hydrolase. As detailed in our response to the reviewer's latest specific comments, we have further investigated the removal of Glu/Asp-mono-ADP-ribose by recombinant PARG in cellular contexts. Our findings reveal that a concentration of PARG almost two orders of magnitude lower than used in the original version of our manuscript is sufficient to completely remove endogenous Glu/Asp-mono-ADP-ribose.

1. The authors should reference a recent paper (Weixler et al., Life Science Alliance, 2022) that described the optimization of detecting ADP-ribosylation in cells. This paper showed that boiling the samples leads to a significant loss of the ADP-ribose signal, consistent with the current study.

We thank the Reviewer for pointing out this oversight. Indeed, we should have included this paper.

We have addressed this oversight by citing the mentioned paper alongside Tachiro et al., JACS, 2023, which was already cited in the original manuscript in another context. Both papers demonstrate that ester-linked ADPr is lost upon boiling, as detailed in the added text in the discussion section: 'Our results are consistent with recent reports showing that boiling the sample causes significant losses of the ADP-ribose signal^{19,54}. While heating at 60°C in a non-denaturing lysis buffer, which requires PARP inhibitor to prevent post-lysis ADPr, has been proposed as an alternative to boiling⁵⁴, a key feature of our western blotting protocol is the avoidance of heating samples. Instead, we maintain them mostly at 4°C and occasionally at room temperature, while still ensuring efficient denaturing cell lysis that inactivates post-lysis activity of PARP1 and PARG. This principle, crucial due to the high thermolability of Asp/Glu-ADPr, allows, to a large extent, the use of standard buffers and procedures, thereby ensuring the protocol can be promptly implemented in any biological laboratory. In contrast, sample preparation for proteomics involves different considerations, as prolonged heating at 37°C is required for effective protein digestion. Hence, for proteomics, we preserved ester-linked ADPr by establishing an acid digestion protocol.'

More specifically, Tachiro et al., JACS, 2023, reported the loss of ADPr signal upon boiling but did not provide an alternative cell lysis protocol for solving this issue. Weixler et al., Life Science Alliance, 2022 avoided boiling by heating the sample at 60°C and adding the PARP inhibitor olaparib to their non-denaturing RIPA-based lysis buffer. This protocol, which has been established mainly in the context of Cys-ADPr, is not ideal for two reasons. First, any heating of the sample could be detrimental to Asp/Glu-ADPr, considering its high thermolability. Second, heating during non-denaturing lysis at 60°C may keep any ADPr modifying enzymes and hydrolases active, as Weixler et al., Life Science Alliance, 2022 specifically showed for PARP1. In contrast, both of these points are overcome by our denaturing protocol, which adheres to the principle of never heating samples. As shown in our new Supplementary Figs. 2a, b, generated in response to Reviewer #1, neither PARP1 nor PARG are active under our conditions, therefore our denaturing protocol does not require the use of inhibitors. This makes it widely applicable to any writer of Asp/Glu-ADPr, and even beyond, to other ester-linked modifications. Therefore, we have also cited Weixler et al., Life Science Alliance, 2022 in the phrase 'Non-denaturing lysis conditions can result in the post-lysis activity of enzymes, including PARP1, requiring the use of inhibitors.'

2. A common strategy to confirm PTM of a target is to mutate the identified sites. Did the authors try to mutate identified glutamates in PARP1?

To address this comment, we used the PARP1E6A construct, in which six glutamate residues in the PARP1 automodification domain (E471, E484, E488, E491, E513, E514) – accounting for approximately 70% of PARP1 ADPr in our cellular proteomic analysis – are mutated to alanine. Intriguingly, when we purified GFP-tagged PARP1E6A from H₂O₂-treated HPF1KO cells and analyzed the automodification status by western blotting, we observed no reduction compared to PARP1WT (new Supplementary Fig. 3e). Mass spectrometry analysis did not detect any ADPr in the automodification domain of the PARP1E6A mutant (new Supplementary Fig. 3f). However, we observed a corresponding increase in ADPr at sites E168, 169, 642. Our data suggest a compensatory mechanism that maintains overall ADPr levels by redirecting enzyme activity to alternative targets within the same protein. This phenomenon is akin to what has been observed for ubiquitination, where, upon mutation of target lysines, E3 ligases are known to ubiquitinate alternative lysines – a behavior documented in multiple studies, such as Bienko et al., Mol Cell 2010; Hou et al., JBC 1994. This finding is particularly insightful, given that the PARP1E6A mutant is commonly used to specifically study the effects of PARP1 automodification on Asp/Glu under the assumption that Asp/Glu-ADPr levels are abolished (Prokhorova et al., Nat Comms, 2021).

We have added the following text 'Intriguingly, mutating all target glutamates present in the automodification domain did not result in a noticeable decrease in total PARP1 mono-ADPr. Mass spectrometry revealed a compensatory increase at other sites, reminiscent of ubiquitination, which often occurs at alternative lysines when target lysines are mutated'

3. The study would be strengthened by showing that Glu-mono-ADP-ribose sites can be identified on proteins beyond the highly abundant PARP1.

We agree and thank the reviewer for this comment, which has motivated us to perform an important experiment that adds considerable depth to our story. We have addressed this comment by performing proteome-wide enrichment of ADPr peptides

using the high-affinity AbD43647 mono-ADPr antibody in HPF1KO cells. The enrichment of ADPr sites generally requires large-scale digestion of milligrams of proteins, which presented a set of unique technical challenges that prompted us to tweak our acidic digestion protocol even further. We realized that due to the high price of Lys-C and Arg-C, a protocol employing both proteases at standard amounts (1:50-1:200 protease:proteins ratio) would be far too expensive for most laboratories, including ours, limiting its adoption and reproducibility. To circumvent this limitation, we tested digestion using only Arg-C and at high dilutions. We observed that we could obtain efficient digestions, as indicated by a low number of miscleavages, even at 1:2000 dilution under acidic conditions (new Supplementary Fig. 4a). Omitting Lys-C and Trypsin has additional advantages for the identification of ADPr peptides. Lack of cleavage on lysine residues generates highly changed, longer peptides that are better suited for ETD fragmentation. Ser-ADPr in particular is likely to benefit since the modified serine is often surrounded by lysine-rich areas such as histone tails.

We then used the peptides generated with our tweaked protocol for large-scale digestion to perform for the first time a peptide pulldown with our high-affinity AbD43647 mono-ADPr antibody. This inaugural application of AbD43647, already popular for western blotting and immunofluorescence, to proteomics marks a significant step, as we expect this high-affinity antibody to gain widespread adoption for immunoprecipitation applications in the coming years. In response to H₂O₂-treatment we identified 203 high confidence sites (localization probability \geq 90%) on 86 protein targets, 151 of which are on Asp/Glu residues (new Fig. 4, Supplementary Figs. 4, 5, and new Supplementary Dataset S2).

This new major addition to our manuscript highlights the abundant and widespread nature of Asp/Glu-ADPr, representing a valuable resource for future studies, and further reinforces the acidic digestion protocol as the ideal proteomic workflow for ADPr.

Given the importance of this point, we have dedicated a new paragraph to it, entitled 'Identification of mono-Asp/Glu-ADPr on additional targets through ester bond preservation', accompanied by a new main figure, two supplementary figures and a supplementary dataset.

4. Removal (albeit incomplete) of Glu/Asp-mono-ADP-ribose on PARP1 requires micromolar concentrations of PARG and several hours. The concentration of PARG in cells is certainly much lower than this. How do the authors reconcile these

discrepancies? Is there an alternative model for the effects of PARG inhibition in cells? For example, could PARG inhibition (or KO) impact the activity of known enzymes that remove Glu/Asp-mono-ADP-ribose?

The reviewer raises several important questions and it is likely that a comprehensive exploration on the regulation and activity of PARG and related enzymes, especially in a cellular context, will require follow-up studies.

First, we would like to highlight that, as pointed out by reviewer #3, our result is supported by a previous study showing complete removal of ADPr from recombinant PARP1 using PARG (Winstall E. et al. Exp Cell Res. 1999). Furthermore, hydrolysis of mono-Asp/Glu-ADPr by recombinant PARG has been recently validated by Tashiro K. et al., J Am Chem Soc 2023, a study we referenced in our original manuscript. Regarding the discrepancy between cellular and *in vitro* data, we note that although PARG is used at high concentrations *in vitro*, the levels of ADPr in recombinant PARP1 used for these reactions are also likely to be dramatically higher than cellular ADPr. Additionally, PARG is directly recruited to sites of DNA damage (Mortusewicz et al., Nuc. Acids Res., 2011), which would increase the local concentration of PARG directly at the modification site.

Nevertheless, we are grateful for the reviewer's comment, as it motivated us to go beyond relying solely on analyzing automodification sites generated *in vitro* by PARP1. Therefore, we attempted to bridge the gap between *in vitro* and cellular data by using recombinant PARG on a source of cellular ADPr, reasoning that it would more closely approximate the conditions in the cellular environment. To this end, we have previously demonstrated that methanol fixation of cells on glass slides keeps ADPr accessible for hydrolytic enzymes, such as ARH3 and snake venom phosphodiesterase (Longarini et al., Mol Cell, 2023) which can then be quantified by immunofluorescence. Treatment of methanol-fixed HPF1KO cells with sub-micromolar concentrations of recombinant PARG, as low as 62 nM, results in the complete removal of mono-Asp/Glu-ADPr signal and only at 31 nM we start to observe incomplete removal (new Fig 6e). Consequently, we have moved the previous Fig. 5e,f of the original manuscript to the Supplementary File (now Supplementary Fig. S6a,d) together with the other *in vitro* PARP1 experiments.

Although our new data help to reconcile the discrepancies between the effects of PARG *in vitro* and in cells, we agree with the reviewer on the importance of considering alternative models. Therefore, we added the sentence 'At the same time, we cannot dismiss the possibility that the observed effects of PARG inhibition on cellular mono-

Asp/Glu-ADPr levels might, at least in part, be attributed to its influence on known enzymes involved in the removal of mono-Asp/Glu-ADPr' to the discussion. Additionally, we cannot exclude the possibility that PARG may not serve as a universal hydrolase for mono-Asp/Glu-ADPr. Instead, it may specifically act on certain targets, with other targets potentially being reversed by different, known hydrolases, such as TARG1.

Reviewer #3 (Remarks to the Author):

This manuscript presented by Longarini and Matic is essentially based on two observations. Firstly, the study rehabilitates PARP-1 in its physiological role of ADP-ribosylation of D/E residues. Secondly, it clarifies the eraser activity of PARG towards the proximal ADP-ribose or the mono-ADP-ribosylation of D/E residues. The study is extremely relevant because these two concepts have been the subject of controversy in the field. The results presented could lead us to reconsider or reinterpret several previous studies. With this in mind, the research theme is more than appropriate and the results obtained will be of interest to a wide range of researchers involved in the study of ADP-ribosylation. On the other hand, the weakness of the manuscript lies in the little attention that the authors make of this controversy and of the previous results presented by other teams which, precisely, support and validate the present observations. It is inconceivable that pioneering studies associated with D/E ADP-ribosylation are not mentioned, discussed, or put into context by the authors. Once again, I mention that this is an original and well-executed study but, in most part, results were expected, particularly in light of previous studies. We would rather have expected a comparison with the numerous ADP-ribosylation sites identified on the D/E residues of PARP-1 both in a cellular context and with recombinant PARP-1. In its current form, the paper is rather technical with a demonstration limited to PARP-1 without proteome-wide application. It remains relevant but perhaps too preliminary for a publication in Nature Communications. Considering that Matic's group and colleagues originally established serine as the primary form of ADP-ribosylation in DNA damage signaling (some even going so far as to speak of artifacts in relation to ADP-ribosylation on D/E

residues), we would have expected a more in-depth context. I am thinking in particular of all the work of Adamietz and Hilz on the lability of ester bonds with ADP-ribose. At that time, the vast majority of studies focusing on the nature of the ADP-ribose linkages provided evidence that ADP-ribosylated histones and PARP-1 itself were primarily modified via hydroxylamine-sensitive ester type of bonds. These explanations are necessary for the reader to understand why the present study is important. Again, Matic and colleagues contributed to establish serine as the primary form of ADP-ribosylation in DNA damage signalling (Palazzo et al. Elife 2018). The manuscript should be restructured and balanced to better appreciate the contribution of each type of modification to the DNA damage response.

We thank the Reviewer for qualifying our study as extremely relevant, original and well-executed and for pointing out that it will of interest to a wide range of researchers interested in ADP-ribosylation. We certainly share the reviewer's view that our manuscript could lead the field to reconsider several previous studies.

We have followed the reviewer's suggestion and provided more in-depth context to the manuscript by discussing and citing the studies mentioned by the reviewer. To some extent the first sentence of our original Results section – 'Historically, glutamate and aspartate were considered the primary acceptors for poly-ADPr in PARP1 signaling' – already reflects the reviewer's comment that 'At that time, the vast majority of studies focusing on the nature of the ADP-ribose linkages provided evidence that ADP-ribosylated histones and PARP-1 itself were primarily modified via hydroxylamine-sensitive ester type of bonds.' Nevertheless, in the introduction, we have expanded the sentence 'In addition, PARP1 alone, as well as virtually all the other members of the PARP family, can attach ADP-ribose to aspartate and glutamate residues' by adding at the end ', which have been historically seen as the primary ADPr targets for histone and PARP1.' and citing the pioneering work of Adamietz and Hilz. This paper is now also cited after the first sentence of the Results section, as mentioned above. Additionally, in the discussion, we have inserted the sentence 'This is despite the longstanding recognition of aspartate and glutamate as primary targets for PARP1-dependent ADPr' following '...the study of more elusive types of ADPr, including on aspartate and glutamate, continues to trail considerably behind'. Following this new sentence we have cited work by Adamietz and Hilz, Poirier, Yu and Kraus. To further underscore the importance of Asp/Glu-ADPr in the DNA damage response, including in the context of Ser-ADPr, we have added the following sentence at the beginning of the discussion 'While our study focuses on mono-ADPr, in terms of amino acid target

specificity, it dispels doubts about the prevalence and significance of Asp/Glu-ADPr in the DNA damage response as demonstrated by prior research, while confirming serine as the primary, but not the only, target in the context of HPF1 signaling.'

We have addressed the reviewer's concern about the application of our approach being limited to PARP1, by applying our high-affinity anti-mono-ADPr for peptide pull-down prior to proteomics for the first time, as detailed in our reply to reviewer #2. These new experiments have demonstrated the applicability of our proteomics approach beyond PARP1 and towards other important targets of mono-Asp/Glu-ADPr. This includes the identification of new Asp/Glu-ADPr targets and sites. Specifically, the combination of the acid digestion protocol with peptide immunoprecipitation based on our high-affinity mono-ADPr antibody, has revealed 18 new Asp/Glu-ADPr targets - most of which known Ser-ADPr targets - and 114 new sites. We wish to underscore that this new experiment, described in an additional main figure, two supplementary figures and a supplementary dataset (Fig. 4; Supplementary Figs. 4, 5; Supplementary Dataset S2), represents a significant addition to our manuscript. It not only broadens the scope of our methodology to previously unknown targets of Asp/Glu-ADPr but also offers a clear path forward for future proteomics studies capable of detecting ADPr on different residues. Also in this context, we have followed the reviewer's suggestion to reference early pioneering studies. Specifically, for the proteomic identification of the H2BE2ADPr site, we cited Ogata et al., JBC 1980, and Burzio et al., JBC 1979.

Major points:

1- The idea of preserving ester bonds by developing a sample preparation pipeline in acidic conditions is original and entirely appropriate. The authors explain that ester bonds are particularly likely to be hydrolyzed in mild basic conditions, hence the idea of carrying out lysis and digestion in acidic conditions. While the concept is certainly applicable to proteomic pipelines, it is unclear how acidic conditions are optimally preserved for detection of ADP-ribosylation by Western blotting. For example, in the "optimal sample preparation" method, cell extracts are loaded with 1X NuPAGE LDS sample buffer. According to the manufacturer, the pH of this solution is 8.5, which is basic enough to hydrolyze very labile ester ADP-ribose conjugates. Technically, all buffers, including gel running buffer and membrane transfer buffer should also be acidic to preserve ester modifications. These conditions are not clearly described. This is crucial as it impacts all blots presented in the manuscript.

We thank the Reviewer for describing our sample preparation protocol in acidic conditions as original and entirely appropriate. The concerns raised by the Reviewer about the pH of the buffers used for western blotting are valid. We agree with the Reviewer that we did not sufficiently explain in the manuscript the complex combination of conditions that result in the lability of ester-linked ADPr.

As shown in our original Figures 2 and 3 along with Supplementary Figures 2 and 3, and independently corroborated by Tashiro K et al. J Am Chem Soc 2023 (referenced as citation 27 in our original manuscript), the lability of Asp-/Glu-ADPr is determined by a combination of factors. These are not limited to the basic pH mentioned by the Reviewer but also include experimental timing and temperature.

Given that efficient protein digestion for proteomics requires incubating samples at 37°C for at least a few hours, and considering that more than 40% of Asp-/Glu-ADPr is lost after one hour incubation at 37°C even at pH 7.4, it was necessary to develop an acid digestion protocol.

In contrast to proteomics, western blotting requires different considerations. We observed the most significant loss of Asp-/Glu-ADPr when boiling, even at neutral pH and for merely 5-10 minutes. Therefore, a major advancement of our manuscript is a protocol that omits boiling step altogether while still achieving efficient, fully denaturing lysis. Consequently, throughout the entire western blotting procedure, it is never necessary to expose the sample to temperatures above room temperature; indeed, most steps are carried out with the sample kept at 4°C. With these temperatures, ester bonds are preserved within the typical pH range of 7 to 7.9, thereby eliminating the need for drastic changes to the western blotting protocol to ensure acidic conditions. Indeed, already in our original manuscript we observed that, at room temperature, this pH range did not lead to noticeable degradation of Asp/Glu-ADPr (previous Supplementary Fig. 2a, now Supplementary Fig. 2c). The simplicity of our western blotting approach - never exceeding room temperature at any step, and rely on standard, widely available buffer keeping the samples at 4°C for most of the procedure - is a significant strength. This ensures easy implementation in any biological laboratory while eliminating the risk of compromising any experimental step by unnecessary acidification of all the buffers.

To directly address the specific concern regarding the NuPAGE LDS sample buffer, we incubated the lysate in 1X LDS sample buffer for different durations. This new experiment performed at room temperature (new supplementary figure 2d)

demonstrates that significant loss of ester-linked ADPr is detected only at 150 and 300 minutes. Marginal loss was observed at 20 and 30 minutes with no noticeable loss at 5 and 10 min. Given that proteins are loaded onto the gel within 5 minutes of mixing with the LDS sample buffer, we can conclude that Asp/Glu ADPr remains stable during this step, namely the loading of cell extracts with 1X NuPAGE LDS sample buffer, as long as the samples are kept at or below room temperature. We have described this new experiment in the text with “Additionally, while long room temperature incubations (≥ 2.5 h) at pH ~ 8.3 lead to substantial loss of Asp/Glu-ADPr, this loss is minimized with short (< 30 min) incubation times (Supplementary Fig. 2d).” This indicates that the lability of Asp/Glu-ADPr is influenced by a combination of factors — specifically, time, pH, and temperature — a conclusion that is corroborated by a recent study’.

For all the SDS-PAGE and western blots in the manuscript, we specifically choose Bis-Tris gels for which, according to the manufacturers, the pH of the transfer buffer ranges from 7.08 to 7.32, and the operating pH during electrophoresis is approximately 7, a result of combining the gel buffer (pH 6.4) and the running buffer (pH 7.3 – 7.7). This is in contrast to the more widespread Tris-Glycine gels, for which the pH of gel buffer is 8.8, and the running and transfer buffer 8.3, which we did not use. Additionally, the buffers are maintained at 4°C and gel electrophoresis is also conducted at 4°C. In this context, it's important to note that Tashiro et al. (J Am Chem Soc 2023) demonstrated no loss of Asp/Glu ADPr at 4°C and pH 7.4, which is slightly higher than the pH used during electrophoresis, even after 16 hours. With our procedure, the overall time of sample processing, from lysis to completion of protein transfer to membranes, is expected to be in the range of 4-5 h. While we have already specified the use of Bis-Tris gels and cold conditions in our original “Immunoblotting” section of the Material and Methods, we now realize that we did not fully explain our choices in this regard.

Therefore, we have expanded the “Immunoblotting” section of the Materials and Methods to comprehensively explain all of our choices with respect to preservation of Asp/Glu-ADPr, including a detailed description of the pH for all buffers used for western blotting.

We have further clarified this point in response to the first suggestion from reviewer #2 by adding the following text to the discussion section: ‘While heating at 60°C in a non-denaturing lysis buffer, which requires PARP inhibitor to prevent post-lysis ADPr, has been proposed as an alternative to boiling⁵⁴, a key feature of our western blotting protocol is the avoidance of heating samples. Instead, we maintain them mostly at 4°C and occasionally at room temperature, while still ensuring efficient denaturing cell lysis

that inactivates post-lysis activity of PARP1 and PARG. This principle, crucial due to the high thermolability of Asp/Glu-ADPr, allows, to a large extent, the use of standard buffers and procedures, thereby ensuring the protocol can be promptly implemented in any biological laboratory. In contrast, sample preparation for proteomics involves different considerations, as prolonged heating at 37°C is required for effective protein digestion. Hence, for proteomics, we preserved ester-linked ADPr by establishing an acid digestion protocol.'

Crucially, we can readily detect Asp-/Glu-ADPr in our samples, except when boiling the sample, as demonstrated in numerous figures throughout the manuscript. Remarkably, this detection is achievable even without utilizing the highest possible amount of our HRP-conjugated anti-mono-ADPr modular antibody. Hence, not only do we lack any theoretical basis to suspect that our straightforward improved western blotting protocol causes detectable loss of Asp-/Glu-ADPr, but more importantly, in practical terms, the preservation of ester-linked modifications achieved is more than adequate for effective detection of Asp-/Glu-ADPr.

2- As shown by the authors, acidic conditions (e.g. 44% formic acid) do not hydrolyze SER-ADP-ribosylation. Therefore, all major ADP-ribosylation linkages should be preserved in acidic conditions (i.e. carboxyl-ester (D/E) and acetal linkages on S/T). However, the authors did not report site-specific SER-ADP-ribosylation in the present study. They explained that HCD fragmentation of Asp/Glu-ADPr peptides leads to the preferential breakage of the diphosphate group of ADP-ribose, leading to the loss of AMP and presence of a phosphoribose-H₂O remnant on the modified amino-acid. This behavior is in contrast to Ser-ADPr peptides, where HCD fragmentation leads largely to complete loss of the modifier, leaving no localization-specific ions. The group of Matic have extensive expertise on the identification of SER-ADP-ribosylation (e.g. by HCD/ETD MS fragmentation) so it is surprising to note that the new method optimized in acidic conditions has not been revalidated via their proteomic platform for SER-ADP-ribosylation. They could have shown that the new technique could be applicable to carry out exhaustive profiling of PARP-1 self-modification. The D/E-specific profiling shown in Fig 3C cannot identify the exact positions of ADP-ribosylation and is therefore unusable in its current form.

Using the raw MS data deposited on ProteomeXchange (PXD048274) we were able to validate the D/E-ADP-ribosylation HCD fragmentation pattern observed by the authors. However, the major SER-ADP-ribosylation sites are located at S499/S504/S508 and not covered by the ARG-C/Lys-C digestion conditions used in this study. The presence of semi-tryptic peptides or partial digestions are necessary to map this region and verify that the HCD fragmentation pattern is truly different between D/E-ADP-ribosylation and SER-ADP-ribosylation. Indeed, the reanalysis of the spectral data by our MS analysis software indicates site-specific ADP-ribosylation on serine residues while clearly observing the presence of the diagnostic ions adenine and AMP. On the other hand, it is difficult to find a tryptic peptide lacking D/E residues to confirm the HCD fragmentation pattern on a specific serine. Validation of the single “acid” approach used for comprehensive identification of all sites should be undertaken.

We are pleased that the reviewer has been able to validate the HCD fragmentation pattern we observed and described in the manuscript.

The reviewer is correct in noting that our acidic digestion approach is suitable for Ser-ADPr, considering that acidic conditions do not hydrolyze Ser-ADPr, as shown in figure 1 of this manuscript. Initially, we did not apply the acidic digestion protocol to Ser-ADPr in our original manuscript, considering such application to be self-evident and of limited novelty, given the efficacy of established proteomics methods for Ser-ADPr. However, we are grateful for the reviewer's comment, as it prompted us to recognize the value of identifying both Ser-ADPr and Asp/Glu-ADPr modification sites on PARP1. In a new experiment, we purified PARP1 from H₂O₂-treated WT cells and successfully identified both Ser-ADPr and Asp/Glu-ADPr modification sites on PARP1 in the same analysis, as anticipated (new Fig. 3e, Supplementary Fig. 3g and Supplementary Dataset S1). Specifically, in the same proteomic analysis utilizing acidic digestion with Arg-C and Lys-C, we have identified ADPr on S499 and S507, which constitute ~90% of the Serine PARP1 automodification (Larsen SC et al., Cell Reports 2018), in addition to the Asp-/Glu-ADPr sites. Please note that, according to several studies (e.g. Bonfiglio JJ et al., Cell 2020; Larsen SC et al., Cell Reports 2018), very little, if any, ADPr occurs on the S504 residue mentioned by the reviewer. We suspect the reviewer may have intended to refer to S507 instead of S508, as the position 508 is occupied by a lysine residue. In the text we have added ‘As expected, given the high stability of Ser-ADPr in acidic conditions (Fig. 1b-d), the acid digestion approach also works for this type of ADPr. Specifically, mass spectrometric analysis of GFP-PARP1 immunoprecipitated

from WT cells enabled the simultaneous detection of both Ser- and Asp/Glu-ADPr sites (new Fig. 3D).'

Regarding the reviewer's comment on the coverage of major Ser-ADPr sites by Arg-C/Lys-C digestion, if we have understood correctly, it is meant that the computational prediction of proteolytic peptides implies that most Ser-ADPr peptides would be too short for identification. However, in practice, trypsin and Arg-C/Lys-C, which in combination have a specificity essentially equivalent to that of trypsin, very inefficiently cleave Arg/Lys in close proximity to ADP-ribosylated Ser, as shown in our previous papers as well as by other groups (e.g. Leidecker O et al. *Nature Chem Biology* 2016; Larsen SC et al., *Cell Reports* 2018). The resulting longer Ser-ADPr peptides can be effectively identified by HCD fragmentation. In our original experiment (old Supplementary Fig. 3d, now Fig. 3c), we identified 176 ms/ms spectra corresponding to the peptide AEPVEVVAPRGKSGAALSK, containing S499, and 46 ms/ms spectra corresponding to KSKGQVKEEGINK, with S507. Additionally, as illustrated in our newly obtained data (new Fig. 4), digesting with Arg-C and without Lys-C is useful for generating longer Lys-rich peptides. This allows better coverage of peptides containing Ser-ADPr sites, which is particularly beneficial for detecting Ser-ADPr site in the PARP1 automodification domain as well as histone tails. Indeed, in our new, proteome-wide experiment, even in HPF1KO, where the levels of Ser-ADPr are dramatically lower but not fully abolished (Hendriks et al., *Nat Comms*, 2021) we could detect Ser-ADPr, for example on H2BS15ADPr as illustrated in the figure below. Shown below are also two additional Ser-ADPr spectra deriving from the new PARP1 pulldown from WT cells, showing that Arg-C+Lys-C digestion can generate peptides spanning the S499 and S507 sites. The full list of Ser-ADPr sites can be found in Supplementary Datasets S1 and S2.

We are not sure if we have fully grasped the essence of the reviewer's concern regarding the differences in HCD fragmentation patterns. The diagnostic ions adenine and AMP consistently appear in MS/MS spectra resulting from HCD fragmentation of both Ser- and Asp/Glu-ADPr peptides. We have demonstrated this for Ser-ADPr in Leidecker et al., *Nat Chem Biol* 2016, and Bonfiglio et al., *Mol Cell* 2017, as well as in the new figure obtained by applying the acid digestion protocol to Ser-ADPr (new Supplementary Fig.3g). Importantly, however, while these diagnostic ions are useful for confirming the presence of ADP-ribose on peptides, they are not useful for localization of the modification to a specific residue within the peptide sequence. What is crucial for establishing the exact site of the modification by HCD is the manner in which ADPr dissociates from the peptide. HCD fragmentation invariably fragments ADP-ribose, generating diagnostic ions in the process. However, the outcome can vary: it may leave a phosphoribose remnant (through the loss of AMP), allowing localization of the modification, or it may dissociate completely from the peptide, which prevents precise localization. The o-glycosidic bond between serine and ADP-ribose is highly labile under HCD, often resulting in the complete dissociation of the modification. As detailed in our short review Bonfiglio et al. 2017 *Nucleic Acid Research*, this lability can lead to misleading localization by the analysis software. The ester bond in Asp/Glu-ADPr is more stable, implying that a phosphoribose remnant (loss of AMP) often remains on the peptide after HCD fragmentation, facilitating the localization of Asp/Glu-ADPr sites. This is the point that we aimed to make in our manuscript, as shown in a representative HCD spectrum (original figure 3e, now Fig. 3d), where the fragment ions b3-AMP and y15-AMP pinpoint the modification on E168. To provide more clarity on this point, we have added '...', rather than the complete loss of the ADP-ribose moiety' after the sentence 'Notably, we observed that HCD fragmentation of Asp/Glu-ADPr peptides causes preferential breakage of the diphosphate group of ADP-ribose'. To further illustrate the presence of phosphoribose-containing fragment ions, we have created a new supplementary figure with representative HCD spectra of Asp/Glu-ADPr peptides and compared to ETD fragmentation of the same site (Supplementary Fig. 5). If the reviewer's concern relates to the potential insufficiency of HCD spectra for precise modification localization despite the presence of the phosphoribose remnant, we would like to highlight that we have also employed ETD, a technique that fully preserves ADPr, thereby unambiguously confirming the exact site of modification. For instance, as illustrated in

the original figure 3e (current Fig. 3d), E168ADPr is confirmed by a high-quality ETD spectrum.

Lastly, regarding the point “*The D/E-specific profiling shown in Fig 3C cannot identify the exact positions of ADP-ribosylation and is therefore unusable in its current form.*”. We have added the position of each identified residue in the plot, and moved this figure to Supplementary Fig. S3d. We have then moved the previous Supplementary Fig. 3d, showing the abundance of PARP1 Glu-ADPr sites in HPF1KO cells, to Fig. 3c with a revised plot to better indicate the modified sites.

3- The authors report ADP-ribosylation sites at E168/169, E190, E471, E488, E491, E513/514 and E642 on PARP-1 in cells. Sites like E190, E488, E491 or E642 were already reported as major ADP-ribosylation sites by the group of Y. Yu (Nature Methods 2013, Cell rep 2017), L. Kraus (Science 2016), G. Poirier (NAR 2012, J Proteome Res 2018), Leung AK (J Proteome Res 2014) and back to Tao et al. (JACS 2009). The authors should explain that the sites they identified had already been validated in cells and with the use of recombinant PARP-1, which validates previous approaches and invalidates the hypothesis of artifactual changes induced by derivatization of the modification as previously reported, notably by the group of ML Nielsen (Hendriks IA et al. Nature Communications 2021).

We have incorporated the reviewer’s suggestion by citing all the recommended papers. To this end, we have added the following sentence to the Results section ‘Our results affirm the validity of previous approaches that have identified, in cellular contexts and with recombinant PARP1, many of the major Asp/Glu-ADPr sites reported here’ immediately following ‘This represents a significant improvement over a recent large-scale proteomics study which did not identify any PARP1 Asp/Glu-ADPr after ADPr enrichment from HPF1KO cells’. The ‘recent large-scale proteomics’ study mentioned refers to Hendriks IA et al. Nature Communications 2021, as cited already in the original manuscript.

4- The schematic model is not really representative of the biochemistry of PARP-1 during DNA damage linked to ADP-ribosylation on residues D/E and S. We do not see the two waves of ADP-ribosylation nor the period where the two modifications could be co-occurring. The authors mention that the formation of mono-Asp/Glu-ADPr is not

limited to HPF1 KO cells but also occurs in WT cells. This should be part of the schematic model.

When we depicted PARP1 without HPF1, the intention was not to imply HPF1 KO cells, but that mono-Asp/Glu-ADPr is formed when PARP1 is not in complex with HPF1 also in WT cells. Nevertheless, we have simplified the schematic model and included the formation of mono-Asp/Glu-ADPr in WT cells, as suggested by the reviewer. If we understood correctly, the reviewer is also suggesting to depict the two waves and co-occurrence of the Asp/Glu-ADPr and Ser-ADPr.

5- The characterization of PARG as a D/E mono-ADP-ribosyl hydrolase is one of the most interesting result of this study, especially the observation that PARG activity towards mono-D/E-ADP-ribosylation is prominent in cells. The authors mentioned that the rapid hydrolysis of PAR by PARG on automodified PARP-1 is followed by a slower and incomplete removal of PARG. They should mention that PARG zymograms performed with ³²P-labeled automodified PARP-1 show a complete disappearance of the radiolabeled moiety (Winstall E. et al. Exp Cell Res. 1999), suggesting a complete removal of the ADP-ribose, which is consistent with the activity observed in cells. Thus, it is possible to observe total erasing depending on the experimental conditions. Zymograms should be performed to support their results.

We thank the reviewer for highlighting our analysis of PARG as a hydrolase of mono-Asp/Glu-ADPr as one of most interesting aspect of our work. As detailed below, we have addressed this by conducting new experiments, incorporating the suggested reference, and making changes to the text.

We have cited the suggested paper in the discussion section, following the addition 'This is supported by evidence from PARG zymograms, which demonstrate the complete disappearance of PARP1 automodification'. Additionally, in the introduction, we have incorporated the citation of Winstall E. et al., Exp Cell Res., 1999, alongside reviewer #1's suggestion to include the original reference 27 earlier in the manuscript. Specifically, we have addressed this by citing both papers and adding the following: 'However, this view has been recently challenged by evidence that recombinant PARG can cleave mono-ADPr from peptides. This is supported by an earlier finding that PARG can completely degrade poly-ADPr, including the initial ADP-ribose moiety', This follows the description stating: 'By contrast, PARG degrades poly-ADPr and,

according to the current consensus, is unable to remove the initial protein-ribose bond, leaving a mono-ADPr remnant on its targets'. Furthermore, we have cited Winstall E. et al., *Exp Cell Res.*, 1999 in the Results sections, by modifying the sentence '...encouraged by a recent report suggesting that recombinant PARG can remove mono-Asp/Glu-ADPr from in vitro modified peptides and by the complete removal of PARP1 automodification observed in PARG zymograms.'

We have explored the feasibility of conducting Zymogram experiments. However, in the process we have realized that preparing the Zymogram, as used in Winstall E. et al. *Exp Cell Res.* 1999, requires extensive incubation times at pH 7.5: specifically, 24 hours at room temperature followed by 3 hours at 37°C. As illustrated in the figure below, these conditions result in a significant loss of Asp/Glu-ADPr, which is consistent with independent observations (Tashiro K. et al., *J Am Chem Soc* 2023). Given our extensive exploration of mono-Asp/Glu-ADPr hydrolysis by PARG using methods that preserve this PTM - including western blotting (previously Fig. 5 and 6c, after revision 6 and 7c) immunofluorescence (previously Fig. 5c, now Fig. 6c) and new mass spectrometric and immunofluorescence analyses (new Fig. 6e and Supplementary Figs. 6e,f) – we consider it unnecessary and somewhat beyond the scope, in light of our manuscript's emphasis on preserving ester bonds, to seek further validation with a technique that compromises Asp/Glu-ADPr integrity.

Instead of conducting a Zymogram experiment, we performed mass spectrometric analysis following our acid digestion protocol for preserving Asp/Glu-ADPr to examine the removal of mono-Asp/Glu-ADPr peptides by recombinant PARG (new Supplementary Fig. 6e,f). Under these conditions, PARG treatment resulted in a ~10-fold reduction in mono-ADPr signal. Additionally, we utilized immunofluorescence to assess the cleavage of cellular mono-Asp/Glu-ADPr at various concentrations of recombinant PARG (new Fig. 6e). We observed complete removal of the mono-ADPr signal even at submicromolar concentrations of PARG. Please also note that our findings regarding the in vitro hydrolysis of mono-Asp/Glu-ADPr by recombinant PARG have been independently validated by Tashiro K. et al., *J Am Chem Soc* 2023, a study we referenced in the original manuscript.

Automodified PARP1EQ

24 h at room temp. + 3 h at 37 °C
(50 mM Na-phosphate pH 7.5, 50 mM NaCl,
10% glycerol, 1 % Triton X-100, 10 mM β -ME)

Minor points:

1- The authors mentioned: Although core histones are abundantly ADP-ribosylated on serine residues during DNA damage, and on glutamate in other biological contexts, we did not observe Asp/Glu-ADPr on histones under our experimental conditions. They should cite Karch KR et al. (Mol Biosyst 2017). Their data suggest that specific D/E residues are present of the nucleosomal surface.

While we already cited Huan D et al., Mol Cell 2020, in the original manuscript in reference to the role of glutamate D/E ADPr in other biological contexts, we agree with the reviewer that citing Karch KR et al., Mol Biosyst 2017, is also necessary. Accordingly, we have included citations for both papers in the revised manuscript. Notably, in the new proteome-wide mono-ADPr enrichment we performed, we were able to identify histone ADPr-ribosylation, particularly on H1 histones, but also on H2B (H2BEADPr, see also the new Fig. 4d and Supplementary Fig. 5a). With respect to the western blot data, low stoichiometry of histone Asp/Glu-ADPr may render these sites hard to detect without ADPr specific enrichment. Therefore, we have also revised the manuscript text by omitting “In contrast to mono-Ser-ADPr, which predominantly targets PARP1 and histones, our findings indicate that mono-Asp/Glu-ADPr is more uniformly distributed across several protein targets (Fig. 2a). Although core histones are abundantly ADP-ribosylated on serine residues during DNA damage and on glutamate in other biological contexts, by western blotting we did not observe mono-Asp/Glu-ADPr on histones under our experimental conditions.” from the text, and included the citations suggested by the reviewer in the new section “Identification of mono-Asp/Glu-ADPr on additional targets through ester bond preservation” following: “We also observed histone Asp/Glu-ADPr (Fig. 4d, Supplementary Fig. 5a, and

Supplementary Dataset S1), confirming that Asp/Glu can also occur on the nucleosomal surface as indicated by previous reports.”

2- In the discussion, the authors imply that the derivatization methods used in MS to identify ADP-ribosylation sites are less valid than the method directly targeting the modification itself. On the other hand, if we consider that the hydroxylamine approach developed by Yu's group to identify ADP-ribosylated D/E residues in cells has essentially been validated by the present approach (major D/E ADP-ribosylation on PARP-1 were identified in addition to thousands of other sites in hundreds of proteins), the two approaches may appear complementary. Moreover, MS signatures generated after derivatization is a common approach in PTM analysis by mass spectrometry. There are obvious advantages to generating diagnostic ions for the analysis of ADP-ribosylated sites but the authors should nevertheless acknowledge the convergence of the two methods.

We thank the reviewer for pointing this out and allowing us to better explain this point that hasn't been addressed adequately in the original manuscript.

We agree with the reviewer that the hydroxylamine proteomics approach is valid, especially since, as the reviewer correctly pointed out, our manuscript confirms several of the main Asp/Glu-ADPr sites identified by this method. We also agree on the advantages offered by detecting ADPr diagnostic ions for mapping ADPr sites. However, we believe that it is crucial to highlight the most important advantage of employing the direct proteomics approach. As demonstrated experimentally in response to the reviewer's second major point, our strategy now enables the identification of ADPr sites on both Ser and Asp/Glu. This is not possible with the hydroxylamine proteomics method, as it inherently excludes the detection of ADPr on residues other than Asp/Glu. Given the multiple amino acid specificities of most PARPs, exemplified by PARP7's modification of both Cys- as well as Asp/Glu, we believe that detecting ADPr on a variety of amino acids within the same proteomics analysis would be more efficient.

To prevent the impression that we consider the hydroxylamine proteomics approach less valid, we changed 'underscored the need for' to 'driven the development of' in the sentence 'The discovery of ADPr's prominence on other amino acids, particularly serine, has driven the development of proteomic approaches capable of detecting

ADPr on targets beyond aspartate and glutamate'. In the same vein, in the discussion we have added the sentence 'While affirming the validity of the hydroxylamine proteomics approach, our approach uniquely enables comprehensive analysis across a spectrum of ADPr forms, including Asp/Glu and the prevalent Ser-ADPr.'

Reviewers' Comments:

Reviewer #1:

Remarks to the Author:

The authors have answered my questions and concerns also including new data in the manuscript. Overall, with the inclusion of additional results the manuscript is further improved and should be of broad interest to the readers of Nature Communications. I recommend accepting it for publication.

Reviewer #2:

Remarks to the Author:

The authors have sufficiently addressed my comments and requested experiments.

Reviewer #3:

Remarks to the Author:

The revised version of the manuscript by Longarini et al represents a remarkable amount of work and responds satisfactorily to all my comments. I reiterate that this study is important for clarifying the biochemistry of PARP-1 ADP-ribosylation and represents an essential evolution of our knowledge in this area. The addition of several validations, including a larger-scale proteomic component, mutants and a revised model, certainly enriches the manuscript. However, I still have a few small points to clarify before its publication in Nature Communications.

1- The title of the manuscript, in itself, implies that glutamate residues are mono-ADP-ribosylated. This is entirely true according to the method used by the authors, including the use of antibodies targeting mono-ADP-ribosylation. Rather, the confusion arises from the fact that poly(ADP-ribosylation) was not mentioned as potential asp/glu modifications, especially at residues E488 and E491. The authors convincingly demonstrate mono-ADP-ribosylation but should mention that poly(ADP-ribosylation) at these same sites is not excluded. A PARylation assay with the PARP1E6 mutant would be interesting to carry out. Since the mutated region containing the six glutamates corresponds to a predominant region in the proteomic analysis (70% of ADP-ribosylation) a modification to the typical smear on gel could be observed even if a compensation phenomenon was observed.

2- The authors mentioned : Our findings provide the first conclusive cellular evidence that HPF1 switches the target specificity of PARP1 from Asp/Glu- to Ser-ADPr, a concept derived from biochemical assays. However, they also state that PARP1 generally targets a conserved set of proteins and its target substrates are therefore largely independent of HPF1. This is with the exception of core histones, which are predominantly Ser-ADP-ribosylated. It might be appropriate here to redefine the concept of "switch" into a much narrower version.

3- A localization probability of >90% identified 267 high-confidence ADP-ribosylation sites. For a study aimed at formally validating the modification of D/E residues, it would be preferable to use a minimum probability of 95%. How does this impact the quality of the overall data set?

4- In supplementary figure 4C, data derived from Larsen et al., Cell Reports 2018, refer to S-ADP-ribosylation (indicated as AspGlu-ADPr intensity in the y-axis).

Thursday, April 18, 2024

Point-by-point answer to reviewers:

NCOMMS-24-02189A (Preserving Ester-linked Modifications Reveals Glutamate and Aspartate mono-ADP-ribosylation by PARP1 and its Reversal by PARG)

Reviewer #1 (Remarks to the Author):

The authors have answered my questions and concerns also including new data in the manuscript. Overall, with the inclusion of additional results the manuscript is further improved and should be of broad interest to the readers of Nature Communications. I recommend accepting it for publication.

We thank the reviewer for recommending acceptance of our manuscript. We are glad that reviewer's questions and concerns have been adequately addressed by our new data.

Reviewer #2 (Remarks to the Author):

The authors have sufficiently addressed my comments and requested experiments.

We thank the reviewer for pointing out that we have sufficiently addressed the comments and requested experiments.

Reviewer #3 (Remarks to the Author):

The revised version of the manuscript by Longarini et al represents a remarkable amount of work and responds satisfactorily to all my comments. I reiterate that this study is important for clarifying the biochemistry of PARP-1 ADP-ribosylation and represents an essential evolution of our knowledge in this area. The addition of several validations, including a larger-scale proteomic component, mutants and a revised model, certainly enriches the manuscript. However, I still have a few small points to clarify before its publication in Nature Communications.

We thank the reviewer for providing a summary of the important additions and for highlighting that the revised manuscript is a remarkable amount of work that has satisfactorily addressed all reviewer's comments. Below we have addressed the few minor points raised by the reviewer at this stage.

1- The title of the manuscript, in itself, implies that glutamate residues are mono-ADP-ribosylated. This is entirely true according to the method used by the authors, including the use of antibodies targeting

mono-ADP-ribosylation. Rather, the confusion arises from the fact that poly(ADP-ribosylation) was not mentioned as potential asp/glu modifications, especially at residues E488 and E491. The authors convincingly demonstrate mono-ADP-ribosylation but should mention that poly(ADP-ribosylation) at these same sites is not excluded. A PARylation assay with the PARP1E6 mutant would be interesting to carry out. Since the mutated region containing the six glutamates corresponds to a predominant region in the proteomic analysis (70% of ADP-ribosylation) a modification to the typical smear on gel could be observed even if a compensation phenomenon was observed.

The reviewer is correct in observing that our manuscript focuses on mono-ADPr. This focus is intentional, as Asp/Glu mono-ADPr represents the novel aspect of our work, whereas poly-ADPr has been extensively studied for several decades. We have clarified this already in the introduction: 'Importantly, while previously identified Asp/Glu-ADPr sites are largely considered as poly-ADPr, typically made undistinguishable from mono-ADPr due to treatments with hydroxylamine, recombinant PARG or PDE, our proteomic approach specifically and unambiguously detects sites of mono-ADPr.' In addition, in the results section, we have already indicated that the identified sites can serve as targets of both mono-ADPr and poly-ADPr with 'Although Asp/Glu residues have traditionally been associated almost exclusively with poly-ADPr, our mono-ADPr proteomics analysis has revealed that these identified sites can also serve as targets for monomeric ADPr.' Please also note that in several figures, we have already explored the impact of our new protocols on poly-ADP-ribosylation smears, using the WWE reagent and CST antibody. Therefore, although it is not its primary focus, our manuscript already explores and discusses poly-ADPr in the context mentioned by the reviewer.

The reviewer is correct in noting that E488 and E491 have been identified by numerous proteomics studies over the past decades. Considering this, along with the specific focus of our manuscript, we believe that further exploration of poly-ADPr is beyond the scope of our current work. Additionally, it is technically not feasible to study the impact of specific mutations on poly-ADPr in the same way as for mono-ADPr, as it is not possible to specifically detect sites of poly-ADPr by mass spectrometry.

2- The authors mentioned : Our findings provide the first conclusive cellular evidence that HPF1 switches the target specificity of PARP1 from Asp/Glu- to Ser-ADPr, a concept derived from biochemical assays. However, they also state that PARP1 generally targets a conserved set of proteins and its target substrates are therefore largely independent of HPF1. This is with the exception of core histones, which are predominantly Ser-ADP-ribosylated. It might be appropriate here to redefine the concept of "switch" into a much narrower version.

This is very valid point and we thank the reviewer for pointing this out. We agree that it is more appropriate to narrow the definition of 'switch'. We have addressed this by adding 'amino acid' before 'target specificity' in the following two sentences: 'Our findings provide the first conclusive cellular evidence that HPF1 switches the amino acid target specificity of PARP1 from Asp/Glu- to Ser-ADPr, a

concept derived from biochemical assays^{10,43}. As an illustration of the significance of our approach in exploring biology, the previous inability to preserve and, therefore, detect Asp/Glu-ADPr led to the puzzling conclusion that HPF1 does not switch the amino acid target specificity of PARP1 from Asp/Glu- to Ser-ADPr³⁴.

3- A localization probability of >90% identified 267 high-confidence ADP-ribosylation sites. For a study aimed at formally validating the modification of D/E residues, it would be preferable to use a minimum probability of 95%. How does this impact the quality of the overall data set?

This would not significantly alter the results, as the vast majority (~85%) of the identified sites with a localization probability of >90% also have a localization probability of >95%. This serves as testimony of the overall high quality of our data. Please note that a localization probability of >90% is already quite stringent and commonly employed in the field, as evidenced by its use in publications from Michael Nielsen's lab. This is noteworthy considering that a less stringent localization probability of >75% is often used in PTM proteomics. On the other hand, we are not aware of any studies with localization probability of >95%. In addition, localization probabilities are already provided for every single identified ADPr site in our supplementary spreadsheets, allowing readers to consider only those sites that have a localization probability of >95%.

4- In supplementary figure 4C, data derived from Larsen et al., Cell Reports 2018, refer to S-ADP-ribosylation (indicated as AspGlu-ADPr intensity in the y-axis).?

We thank the reviewer for pointing out this oversight in Supplementary Figure 4C. We have corrected the error accordingly. Indeed, the data derived from Larsen et al., Cell Reports 2018, refer to Ser-ADP-ribosylation and not Asp/Glu-ADP-ribosylation, as was mistakenly indicated on the y-axis.